# Single Image Reflection Separation via Dual-Stream Interactive Transformers

**Qiming Hu, Hainuo Wang, Xiaojie Guo**[*]
College of Intelligence and Computing, Tianjin University, Tianjin 300350, China
`huqiming@tju.edu.cn` `hainuo@tju.edu.cn` `xj.max.guo@gmail.com`

## Abstract

Despite satisfactory results on "easy" cases of single image reflection separation, prior dual-stream methods still suffer from considerable performance degradation when facing complex ones, *i.e.*, the transmission layer is densely entangled with the reflection having a wide distribution of spatial intensity. The main reasons come from the lack of concern on the feature correlation during interaction, and the limited receptive field. To remedy these deficiencies, this paper presents a Dual-Stream Interactive Transformer (DSIT) design. Specifically, we devise a dual-attention interactive structure that embraces a dual-stream self-attention and a layer-aware dual-stream cross-attention mechanism to simultaneously capture intra-layer and inter-layer feature correlations. Meanwhile, the introduction of attention mechanisms can also mitigate the receptive field limitation. We modulate single-stream pre-trained Transformer embeddings with dual-stream convolutional features through cross-architecture interactions to provide richer semantic priors, thereby further relieving the ill-posedness of the problem. Extensive experimental results reveal the merits of the proposed DSIT over other state-of-the-art alternatives. Our code is publicly available at https://github.com/mingcv/DSIT.

## 1 Introduction

When images are captured through glass-like mediums (semi-reflectors), the reflected scenes appear together with the transmitted ones to different degrees, influenced by many factors such as the material of medium and the illumination of both scenes, among others [42, 62]. This phenomenon poses significant challenges in various fields, like multi-view stereo imaging, mobile photography, security surveillance, and autonomous driving [49, 43]. Therefore, successfully separating the superimposed layers can, on the one hand, enhance the capability of models to serve downstream applications. On the other hand, it paves the way for tackling a broader spectrum of layer-decomposition tasks, such as image denoising and watermark/obstacle removal [51, 20].

As a long-standing blind source separation problem, single image reflection separation (SIRS) has always been challenging, due to the severe ill-posedness of disentangling two natural image signals. Generally, the superimposed images $\mathbf{I}$ can be formulated as follows:

$$\mathbf{I} = \mathbf{T} + \mathbf{R} + \mathcal{C}(\mathbf{T}, \mathbf{R}). \tag{1}$$

It consists of an additive combination of target transmission and reflection layers ($\mathbf{T}$ and $\mathbf{R}$, respectively) and a residual component, denoted by the mapping $\mathcal{C}$ of the two layers. Note that $\mathcal{C}$ is used to describe the non-linear/linear attenuation of the two layers, thus representing a group of reflection models [23]. In the literature, two main routes of approaches have been delivered. One tendency (single-stream) is to treat the reflection layer as noise/degradation, merely modeling the transmission layer. Alternatively, the other rising trend (dual-stream) pays attention to the reconstruction quality

---

[*]Corresponding Author

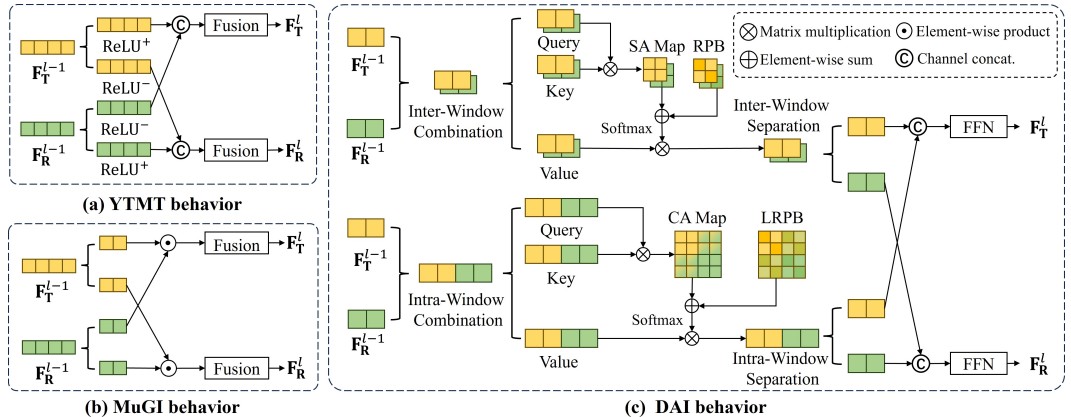

Figure 1: Schematic illustration of dual-stream interactive behaviors, including YTMT [22], MuGI [23], and our proposed Dual-Attention Interaction (DAI) mechanisms, where $\mathbf{F_T}$ and $\mathbf{F_R}$ represent the feature flows of transmission layer and reflection layer respectively. The superscript $l$ of the feature flows denotes the number of building blocks traversed to derive the flows.

of both the transmission and reflection layers. This work follows the latter principle because the reflection layer may also contain valuable information [43] and imposing constraints from both perspectives can better regularize the decomposition [22, 23].

The dual-stream schemes, with IBCLN [29], YTMT [22], and DSRNet [23] as representatives, attempt to estimate both of the two layers with Siamese networks that employ two sub-networks with identical architecture and shared weights. Particularly, YTMT and DSRNet advocated dual-stream feature interactions to facilitate the information flow between the streams and finally reconstruct the decoupled layers. Though being effective, the interaction mechanisms used in these two methods do NOT explicitly assess the correlation between dual-stream features during interactions, as illustrated in Fig. 1 (a) and (b). More concretely, assuming we have dual-stream features at a certain stage in the networks, previous interaction strategies directly pass the undesired information at the current stage (may be required again at subsequent stages) from one branch to the other without checking if the passed information is needed by the sibling. Actually, in intermediate blocks, some information is very likely delivered back and forth, making the separation process ineffective and inefficient. In contrast, the attention mechanisms in Transformers assign small weights to token pairs with low similarities, which are further suppressed by the Softmax function. In other words, employing a cross-attention mechanism seems to have the potential for improving dual-stream interaction. Furthermore, due to the presence of correlated scenes in reflection superposition phenomena across entire images, the task demands a keen perception of global information in both streams. Motivated by the above analysis, we propose a strategy, called dual-attention interaction, consisting of a dual-stream self-attention and a dual-stream cross-attention, as depicted in Fig. 1 (c), to extract both the intra-layer and inter-layer feature correlations explicitly.

In addition, most of recent state-of-the-art methods [60, 48, 22, 23] adopted networks (*e.g.*, Hyper-Column [60, 48, 22] and Feature Pyramid [23]) pre-trained on high-semantic tasks to assist feature extraction. In this work, we argue that Transformers, thanks to their generalization and selective attention characteristics, should be more powerful to the target task. However, pre-trained Transformer models typically have trouble in dense prediction due to the lack of inductive biases [8]. To make Transformers suitable, we develop a Dual-Architecture Interactive Encoder (DAIE), which enables the interaction between semantically rich features extracted by a pre-trained Transformer and local dual-stream features extracted by a CNN. By this means, the Cross-Architecture Interactions (CAI) can balance the global and local perspectives and combine high-semantic priors with the low-semantic demands of the reflection separation.

In summary, our primary contributions are as follows:

- We propose a novel Dual-Attention Interaction (DAI) mechanism to energize Dual-Stream Interactive Transformers. DAI introduces the explicit correlation assessment within dual streams to effectively address the challenge of reflection separation;

- We customize a bridge, namely the Dual-Architecture Interactive Encoder (DAIE), to connect the pre-trained Transformer model with the task of layer decomposition, which alleviates the inherent ill-posedness of the problem;

- Through extensive experiments on multiple datasets, we demonstrate the efficacy of our design with superior performance over other SOTA competitors, both quantitatively and qualitatively. Moreover, the better generalizability compared to previous methods is also verified.

## 2 Related Work

**Low-level Vision Transformers**. Building upon the attention mechanism [3], Transformers were initially developed by the community of natural language processing [41], and soon became popular across various domains because of their remarkable modeling [2, 17], scaling [24, 56, 36], and transferring [10, 6, 4] abilities. Introduced by [12], Vision Transformers (ViTs) have shown advantages on a large number of visual tasks [56, 34, 66, 57, 50, 8].

For low-level purposes, IPT [6] was developed to handle multiple restoration tasks with a shared standard Transformer body, which required a large number of parameters for good performance without suitable task-related priors. Swin Transformer [35] introduced the shifted window attention mechanism, which reduced the computational cost of attention while incorporating inductive biases for images, inspiring a series of subsequent works. SwinIR [33] equipped residual-in-residual structures [46, 61] with the Swin Transformer block, exploring its capability in low-level vision tasks. ELAN [59] performed multiple window attentions of varying sizes in parallel and fused them, using grouped four-directional offset convolution layers for local feature extraction and cross-window association. Chen *et al.* [7] introduced an overlapping attention mechanism in their HAT model to establish cross-window connections and employed same-task pre-training for better performance. Zhang *et al.* proposed ART [58], which utilized a sparse window attention mechanism akin to dilated convolution, alternating it with window attention, thereby replacing the shifted window mechanism. UFormer [47] embedded window attention modules into a U-shaped network, which captured cross-window associations beyond the current scale. Moreover, Restormer [55] introduced a transposed attention mechanism, resembling channel attention. While faster, this approach somewhat neglected spatial correlations. Retinexformer [5] employed this design for low-light image enhancement. DAT [9] alternated between window self-attention and window transposed self-attention to address spatial correlation deficiencies. *Overall, these methods mostly opt to validate their designs on tasks like image super-resolution, which focus on reconstructing a single component, overlooking the intrinsic advantages of attention mechanisms in component decomposition tasks.*

**Single Image Reflection Separation**. Single image reflection separation, with looser data assumptions, relies more heavily on priors to alleviate its inherent ill-posedness. Traditional methods developed priors like edge sparsity [28, 27], manual annotation [26], or relative smoothness [31], some of which were further leveraged by deep learning methods. Although multiple-image solutions [37, 1, 16, 39, 30, 18, 38, 51, 53, 19] have shown satisfactory performance facing weaker ill-posedness, *these methods typically rely on sequences of images captured with rotating polarizers or moving cameras, which limits their applicability.*

In deep learning methods, CEILNet [14] applied the relative smoothness assumption to data synthesis and used an edge detection network to emphasize edge information. Zhang *et al.* [60] proposed a gradient mutual exclusion loss to promote edge sparsity and introduced the HyperColumn and perceptual loss to incorporate high-level semantics. ERRNet [48] aimed to expand the receptive field and utilized non-aligned images for data augmentation. *These methods employed end-to-end single-branch networks for estimation, but the lack of interaction between layers led to inefficiency and untidy separations.* In another way, BDN [54] alternated between estimating transmission and reflection layers, considering their mutual dependency. RAGNet [32] estimated reflection first and then used its features to modulate the estimation of the transmission layer and mask. Dong *et al.* [11] emphasized the importance of the reflection layer, using multi-scale Laplacian features with LSTM for iterative estimation. RRW [25] utilized a cascaded reflection detector and remover and proposed a more reasonable data acquisition scheme. Song *et al.* [40] proposed a robust SIRR model based on a multi-scale Transformer architecture, but it only learned to restore the transmission layer, and the usage of the Transformer was not well motivated. IBCLN [29] introduced a convolutional LSTM network, using a dual-branch structure for reflection and transmission reconstruction but lacking interaction between branches for cross-verifying the accuracy of decoupling. Hu and Guo proposed a dual-stream

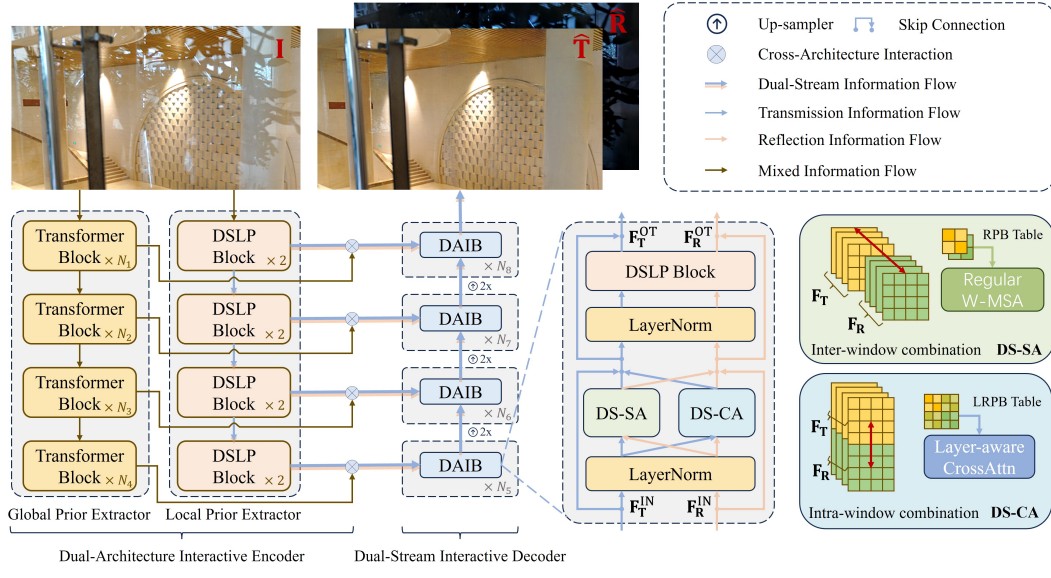

Figure 2: (a) The overall architecture of our proposed Dual-Stream Interactive Transformer, which consists of a dual-architecture interactive encoder and a dual-stream interactive decoder, injecting the global prior into local features and aggregating them in dual-stream from bottom to up. (b) A visual illustration of our proposed dual-attention interactive block, which provides both intra-layer self-attention and inter-layer cross-attention, capturing holistic feature correlations.

interactive reflection separation strategy [22], called YTMT, assessing information utility through activation functions and exchanging low-value information, facilitating information retention and efficient interaction in the high-dimensional feature space. Their subsequent DSRNet [23] used MuGI for efficient interaction and introduced a non-linear residual component to estimate the overexposure during the layer superimposition, achieving state-of-the-art performance. *However, neither the YTMT nor the MuGI mechanism explicitly assessed the correlation of exchanged information, which can introduce errors and lower the decoupling efficiency.* To further alleviate the ill-posedness of the problem, a recent concurrent work [44] utilized wavelet priors and diffusion models to guide frequency-domain-based reflection removal. The other work [64] used natural language prompts for reflection separation, which, however, required accurate paired prompts, incurring additional costs.

## 3 Methodology

Our overall architecture is illustrated in Fig. 2 (a), which comprises a Dual-Architecture Interactive Encoder (DAIE) and a Dual-Stream Interactive Decoder (DSID). The DAIE leverages both a pretrained Transformer (Global Prior Extractor, GPE) and a task-specific dual-stream CNN (Local Prior Extractor, LPE), capturing global and local features through specialized extractors. The mixed global information is then injected into the dual-stream local flows via Cross-Architecture Interactions (CAI), ensuring comprehensive information utilization. Subsequently, the DSID separates and aggregates the embeddings hierarchically through our newly developed Dual-Attention Interactive Block (DAIB, illustrated in Fig. 2 (b)). These components are detailed in the following subsections.

### 3.1 Dual-Attention Interactive Block

As depicted in Fig. 2 (b), our proposed DAIB embraces a dual-stream design, taking both transmission and reflection feature flows ($\mathbf{F}_\mathbf{T}^{\mathrm{IN}}$ and $\mathbf{F}_\mathbf{R}^{\mathrm{IN}}$, respectively) as inputs. After that, a layer normalization and two parallel attention mechanisms, namely dual-stream self-attention (DS-SA) and dual-stream cross-attention (DS-CA) are applied to the feature flows, capturing both inter- and intra-layer correlations. Subsequently, we derive the output feature flows $\mathbf{F}_\mathbf{T}^{\mathrm{OT}}$ and $\mathbf{F}_\mathbf{R}^{\mathrm{OT}}$ after passing the features through a layer normalization and a feed-forward network in the form of the dual-stream locality-preserving

block (DSLP Block). The detailed computation procedure is displayed in Alg. 1. We provide a detailed explanation of the dual attention mechanism in the remainder of this subsection.

**Efficient Dual-Stream Cross-Attention Mechanism**. We present a simple yet effective cross-attention mechanism for dual-stream Transformer models via an extension of the self-attention mechanism. Given the feature streams of transmission layer $\mathbf{F_T} \in \mathbb{R}^{N \times C}$ and reflection layer $\mathbf{F_R} \in \mathbb{R}^{N \times C}$, we concatenate them along the token dimension to form the input matrix $\mathbf{X}_{\mathrm{CA}} \in \mathbb{R}^{2N \times C} = \begin{bmatrix} \mathbf{F_T} \\ \mathbf{F_R} \end{bmatrix}$. We then compute the query $\mathbf{Q}_{\mathrm{CA}}$, key $\mathbf{K}_{\mathrm{CA}}$, and value $\mathbf{V}_{\mathrm{CA}}$ matrices for cross-attention by applying the linear transformations:

$$\mathbf{Q}_{\mathrm{CA}} = \mathbf{X}_{\mathrm{CA}}\mathbf{W}_q, \quad \mathbf{K}_{\mathrm{CA}} = \mathbf{X}_{\mathrm{CA}}\mathbf{W}_k, \quad \mathbf{V}_{\mathrm{CA}} = \mathbf{X}_{\mathrm{CA}}\mathbf{W}_v, \tag{2}$$

where $\mathbf{W}_q, \mathbf{W}_k, \mathbf{W}_v \in \mathbb{R}^{C \times D}$ denote the weight matrices that project the input features from $C$-dimensional channels into $D$-dimensional hidden representations. The cross-attention score matrix $\mathbf{A}_{\mathrm{CA}} \in \mathbb{R}^{2N \times 2N}$ are computed as:

$$\begin{aligned} \mathbf{A}_{\mathrm{CA}} \in \mathbb{R}^{2N \times 2N} &= \mathrm{Softmax}(\mathbf{Q}_{\mathrm{CA}}\mathbf{K}_{\mathrm{CA}}^{\top}) = \mathrm{Softmax}(\begin{bmatrix} \mathbf{F_T} \\ \mathbf{F_R} \end{bmatrix} \mathbf{W}_q \mathbf{W}_k^{\top} \begin{bmatrix} \mathbf{F_T^{\top}} & \mathbf{F_R^{\top}} \end{bmatrix}) \\ &= \mathrm{Softmax}(\begin{bmatrix} \mathbf{F_T^{\top}}\mathbf{W}_q\mathbf{W}_k^{\top}\mathbf{F_T^{\top}} & \mathbf{F_T^{\top}}\mathbf{W}_q\mathbf{W}_k^{\top}\mathbf{F_R^{\top}} \\ \mathbf{F_R^{\top}}\mathbf{W}_q\mathbf{W}_k^{\top}\mathbf{F_T^{\top}} & \mathbf{F_R^{\top}}\mathbf{W}_q\mathbf{W}_k^{\top}\mathbf{F_R^{\top}} \end{bmatrix}), \end{aligned} \tag{3}$$

where the intra-layer terms $\mathbf{F_T}\mathbf{W}_q\mathbf{W}_k^{\top}\mathbf{F_T^{\top}}$ and $\mathbf{F_R}\mathbf{W}_q\mathbf{W}_k^{\top}\mathbf{F_R^{\top}}$ represent interactions within the transmission stream $\mathbf{F_T}$ and the reflection stream $\mathbf{F_R}$, respectively. The inter-layer terms $\mathbf{F_T}\mathbf{W}_q\mathbf{W}_k^{\top}\mathbf{F_R^{\top}}$ and $\mathbf{F_R}\mathbf{W}_q\mathbf{W}_k^{\top}\mathbf{F_T^{\top}}$ indicate interactions between $\mathbf{F_T}$ and $\mathbf{F_R}$. By denoting the Softmax function with a scaling factor $\frac{1}{\sqrt{D}}$ as $\mathcal{S}(\cdot)$, the output matrix $\mathbf{Y}_{\mathrm{CA}}$ is then calculated as:

$$\mathbf{Y}_{\mathrm{CA}} = \mathbf{A}_{\mathrm{CA}}\mathbf{V}_{\mathrm{CA}} = \begin{bmatrix} \mathcal{S}(\mathbf{F_T}\mathbf{W}_q\mathbf{W}_k^{\top}\mathbf{F_T^{\top}})\mathbf{F_T}\mathbf{W}_v + \mathcal{S}(\mathbf{F_T}\mathbf{W}_q\mathbf{W}_k^{\top}\mathbf{F_R^{\top}})\mathbf{F_R}\mathbf{W}_v \\ \mathcal{S}(\mathbf{F_R}\mathbf{W}_q\mathbf{W}_k^{\top}\mathbf{F_T^{\top}})\mathbf{F_T}\mathbf{W}_v + \mathcal{S}(\mathbf{F_R}\mathbf{W}_q\mathbf{W}_k^{\top}\mathbf{F_R^{\top}})\mathbf{F_R}\mathbf{W}_v \end{bmatrix}. \tag{4}$$

We further simplify the form of $\mathbf{Y}_{\mathrm{CA}}$ by introducing $\mathcal{G}(\mathbf{Z}_1, \mathbf{Z}_2) = \mathcal{S}(\mathbf{Z}_1\mathbf{W}_q\mathbf{W}_k^T\mathbf{Z}_2^{\top})\mathbf{Z}_2\mathbf{W}_v$, where $\mathbf{Z}_1 \in \mathbb{R}^{N \times C}$ and $\mathbf{Z}_2 \in \mathbb{R}^{N \times C}$ can be chosen between $\mathbf{F_T}$ and $\mathbf{F_R}$, yielding the follows:

$$\mathbf{Y}_{\mathrm{CA}} = \begin{bmatrix} \mathcal{G}(\mathbf{F_T}, \mathbf{F_T}) + \mathcal{G}(\mathbf{F_T}, \mathbf{F_R}) \\ \mathcal{G}(\mathbf{F_R}, \mathbf{F_T}) + \mathcal{G}(\mathbf{F_R}, \mathbf{F_R}) \end{bmatrix} = \begin{bmatrix} \mathbf{F_T^{\mathrm{CA}}} \\ \mathbf{F_R^{\mathrm{CA}}} \end{bmatrix}. \tag{5}$$

We finally obtain the output of the dual-stream cross-attention as $\mathbf{F_T^{\mathrm{CA}}} = \mathcal{G}(\mathbf{F_T}, \mathbf{F_T}) + \mathcal{G}(\mathbf{F_T}, \mathbf{F_R})$ and $\mathbf{F_R^{\mathrm{CA}}} = \mathcal{G}(\mathbf{F_R}, \mathbf{F_T}) + \mathcal{G}(\mathbf{F_R}, \mathbf{F_R})$, which are the combined effects of intra-layer and inter-layer interactions. Meanwhile, if we concatenate the dual-stream features along the batch dimension, obtaining the input matrix for the dual-stream self-attention mechanism $\mathbf{X}_{\mathrm{SA}} \in \mathbb{R}^{2 \times N \times C}$, we can further boost the parallelism of our model.

**Dual-Attention Design**. Based on the above analysis, we can define the following dual-attention mechanism:

$$\begin{cases} \mathbf{Y}_{\mathrm{SA}} = \mathrm{DS\text{-}SA}(\mathbf{Q}_{\mathrm{SA}}, \mathbf{K}_{\mathrm{SA}}, \mathbf{V}_{\mathrm{SA}}) = \mathrm{SoftMax}(\mathbf{Q}_{\mathrm{SA}}\mathbf{K}_{\mathrm{SA}}^{\top}/\sqrt{D} + \mathbf{B}_{\mathrm{SA}})\mathbf{V}_{\mathrm{SA}}, \\ \mathbf{Y}_{\mathrm{CA}} = \mathrm{DS\text{-}CA}(\mathbf{Q}_{\mathrm{CA}}, \mathbf{K}_{\mathrm{CA}}, \mathbf{V}_{\mathrm{CA}}) = \mathrm{SoftMax}(\mathbf{Q}_{\mathrm{CA}}\mathbf{K}_{\mathrm{CA}}^{\top}/\sqrt{D} + \mathbf{B}_{\mathrm{CA}})\mathbf{V}_{\mathrm{CA}}, \end{cases} \tag{6}$$

where $\mathbf{Q}_{\mathrm{SA}}$, $\mathbf{K}_{\mathrm{SA}}$, and $\mathbf{V}_{\mathrm{SA}}$ are derived as in the DS-CA. Note that, the number of tokens is doubled in DS-CA compared to DS-SA. To reduce the computation burden, we employ a window-based attention mechanism for our dual-attention design. In this way, $\mathbf{Q}_{\mathrm{SA}}, \mathbf{K}_{\mathrm{SA}}, \mathbf{V}_{\mathrm{SA}} \in \mathbb{R}^{2N_T \times N_W \times D}$, $\mathbf{Q}_{\mathrm{CA}}, \mathbf{K}_{\mathrm{CA}}, \mathbf{V}_{\mathrm{CA}} \in \mathbb{R}^{N_T \times 2N_W \times D}$, where $N_T$ denotes the total number of windows, $N_W$ stands for the window size. $\mathbf{B}_{\mathrm{SA}} \in \mathbb{R}^{N_W \times N_W}$ represents the relative position bias [35], which provides the same bias values with respect to the same distance between two tokens in a window. It is obtained by indexing a learnable lookup table $\mathbf{B}_{\mathrm{SA}}^{\mathrm{LUT}} \in \mathbb{R}^{(2\sqrt{N_W}-1) \times (2\sqrt{N_W}-1)}$ through the predefined relative indexes $\mathbf{U}_{\mathrm{SA}} \in \mathbb{R}^{N_W \times N_W}$. Each item of $\mathbf{U}_{\mathrm{SA}}$ is a mapped distance of two locations: $u_{ij} = t(p_i - p_j)$. $p$ is a 2-D point in a window, the coordinates of which fall between 0 and $\sqrt{N_W} - 1$, and each

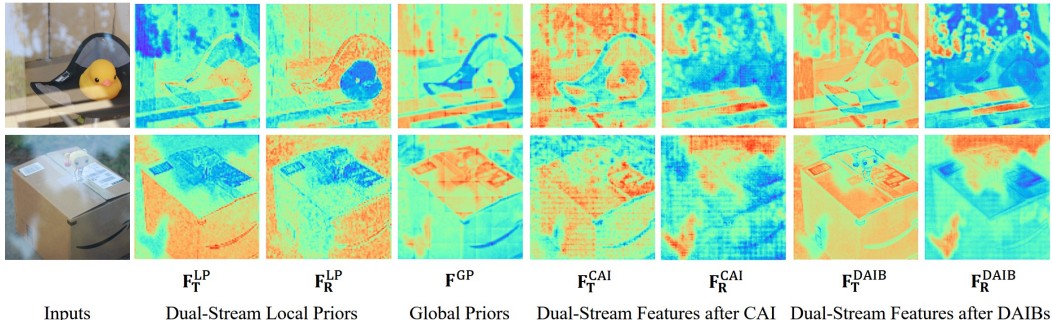

| $\mathbf{F_T^{LP}}$ | $\mathbf{F_R^{LP}}$ | $\mathbf{F^{GP}}$ | $\mathbf{F_T^{CAI}}$ | $\mathbf{F_R^{CAI}}$ | $\mathbf{F_T^{DAIB}}$ | $\mathbf{F_R^{DAIB}}$ |

Inputs     Dual-Stream Local Priors     Global Priors     Dual-Stream Features after CAI     Dual-Stream Features after DAIBs

Figure 3: Visualization of extracted local priors, global priors, their cross-architecture-interacted dual-stream features and features after the DAIBs of two reflection-superimposed inputs. All the above features are from the second level of our DSIT model and are channel-wise averaged to display.

distinct $p_i - p_j$ is mapped into a single index by $t(\cdot)$. For DS-CA, we propose the Layered Relative Position Biases (LRPB), $\mathbf{B}_{\text{CA}} \in \mathbb{R}^{2N_W \times 2N_W}$, which are indexed from the extended lookup table $\mathbf{B}_{\text{CA}}^{\text{LUT}} \in \mathbb{R}^{(2\sqrt{N_W}-1) \times (2\sqrt{N_W}-1) \times 3}$ by the layered relative indexes $\mathbf{U}_{\text{CA}} \in \mathbb{R}^{2N_W \times 2N_W}$. Each element is mapped by subtracting two 3-D points $u'_{ij} = t'(v_i - v_j)$. $v$ is a 3-D point in a layered window, with an additional dimension representing to which layer the token belongs, and $t'(\cdot)$ maps 3-D locations into single indexes.

**Dual-Stream Locality-Preserving Block**. Since reflection separation is a dense prediction task, a primary consideration during architecture design is to maintain the local information. Therefore, we introduce the DSLP Block in our DSIT structure, which can be *any convolutional dual-stream* network modules. To focus on the enhanced interaction capabilities achieved by our dual-attention design, we avoid introducing additional novel local modules, opting instead to employ the MuGI Block [23] as the implementation of the DSLP Block. This approach isolates the performance gains attributed solely to the dual-attention mechanism, as evidenced in comparisons with models like DSRNet. One could, of course, substitute our design with alternative specialized dual-stream modules, potentially achieving even better model performance.

## 3.2 Dual-Architecture Interactive Encoder

As depicted in Fig. 2 (a), our proposed DAIE integrates both global and local prior extractors. The single-stream global features modulate the dual-stream local features hierarchically through cross-architecture interactions (CAI), which are implemented with our proposed dual-attention interactive blocks. Formally, we have DAIB($\mathbf{F}^{\text{GP}}, \mathbf{F}_{\mathbf{T}}^{\text{LP}}$) and DAIB($\mathbf{F}^{\text{GP}}, \mathbf{F}_{\mathbf{R}}^{\text{LP}}$), where $\mathbf{F}^{\text{GP}}$ and $\mathbf{F}_{\mathbf{T}}^{\text{LP}}, \mathbf{F}_{\mathbf{R}}^{\text{LP}}$ represent global and local features respectively. $\mathbf{F}_{\mathbf{T}}^{\text{LP}}$ denotes the transmission information flow and $\mathbf{F}_{\mathbf{R}}^{\text{LP}}$ signifies the reflection stream. In an effort to provide an intuitive understanding of our DAIE design, we illustrate the feature visualization of DSIT in Fig. 3. As shown, the dual-stream local priors focus on different components of the inputs but lack precise layer-specific attention. After being modulated by the global priors $\mathbf{F}^{\text{GP}}$ via our proposed CAI and aggregated with the lower stream, we obtain $\mathbf{F}_{\mathbf{T}}^{\text{CAI}}$ and $\mathbf{F}_{\mathbf{R}}^{\text{CAI}}$, which are significantly separated. Furthermore, the modulated dual-stream features are fed into a group of DAIBs, resulting in feature separations $\mathbf{F}_{\mathbf{T}}^{\text{DAIB}}, \mathbf{F}_{\mathbf{R}}^{\text{DAIB}}$ of higher quality.

## 3.3 Loss Function

**Pixel reconstruction loss.** To compel the consistency of the restored layers $\hat{\mathbf{T}}$ and $\hat{\mathbf{R}}$ in the spatial domain with their ground-truth scenes $\mathbf{T}$ and $\mathbf{R}$ and the layer superimposition modeling, we introduce the following loss function:

$$\mathcal{L}_{pix} := \|\hat{\mathbf{T}} - \mathbf{T}\|_2^2 + \|\hat{\mathbf{R}} - \mathbf{R}\|_2^2 + \alpha\|\mathbf{I} - (\hat{\mathbf{T}} + \hat{\mathbf{R}}) - \mathcal{E}(\hat{\mathbf{T}}, \hat{\mathbf{R}})\|_1, \tag{7}$$

where $\mathcal{E}$ denotes a learnable term to constitute the reflection superposition. $\|\cdot\|_2$ and $\|\cdot\|_1$ represent the $\ell_2$ and $\ell_1$ norms, respectively. $\alpha$ is a hyperparameter to balance the intra-layer and inter-layer fidelity. By enforcing the reconstruction loss with a learnable residual term, the restored layers appear to be cleaner and completed.

Table 1: Quantitative results on four real-world testing datasets of SIRS models. The best results are displayed in **bold**, while the second-best are underlined. † means data setting II is employed to train the model. * represents additional prompts are introduced. △ reflects extra data pairs are involved.

| Methods | Real20 (20) | | Objects (200) | | Postcard (199) | | Wild (55) | | Average | |
|---|---|---|---|---|---|---|---|---|---|---|
| | PSNR | SSIM | PSNR | SSIM | PSNR | SSIM | PSNR | SSIM | PSNR | SSIM |
| Zhang et al. [60] | 22.55 | 0.788 | 22.68 | 0.879 | 16.81 | 0.797 | 21.52 | 0.832 | 20.08 | 0.835 |
| BDN [54] | 18.41 | 0.726 | 22.72 | 0.856 | 20.71 | 0.859 | 22.36 | 0.830 | 21.65 | 0.849 |
| ERRNet [48] | 22.89 | 0.803 | 24.87 | 0.896 | 22.04 | 0.876 | 24.25 | 0.853 | 23.53 | 0.879 |
| IBCLN [29] | 21.86 | 0.762 | 24.87 | 0.893 | 23.39 | 0.875 | 24.71 | 0.886 | 24.10 | 0.879 |
| RAGNet [32] | 22.95 | 0.793 | 26.15 | 0.903 | 23.67 | 0.879 | 25.53 | 0.880 | 24.90 | 0.886 |
| DMGN [15] | 20.71 | 0.770 | 24.98 | 0.899 | 22.92 | 0.877 | 23.81 | 0.835 | 23.80 | 0.877 |
| Zheng et al. [63] | 20.17 | 0.755 | 25.20 | 0.880 | 23.26 | 0.905 | 25.39 | 0.878 | 24.19 | 0.885 |
| YTMT [22] | 23.26 | 0.806 | 24.87 | 0.896 | 22.91 | 0.884 | 25.48 | 0.890 | 24.05 | 0.886 |
| RobustSIRR [40] | 23.30 | 0.827 | 24.90 | 0.917 | 19.91 | 0.868 | 23.67 | 0.884 | 22.59 | 0.889 |
| DSRNet [23] | 24.23 | 0.820 | 26.28 | 0.914 | 24.56 | 0.908 | 25.68 | 0.896 | 25.40 | 0.905 |
| PromptRR [44] | 24.11 | 0.813 | 24.17 | 0.859 | 23.03 | 0.895 | 26.43 | **0.930** | 23.95 | 0.880 |
| Ours | **25.06** | **0.836** | **26.81** | **0.919** | **25.63** | **0.924** | **27.06** | 0.910 | **26.27** | **0.917** |
| Dong et al.† [11] | 23.34 | 0.812 | 24.36 | 0.898 | 23.72 | 0.903 | 25.73 | 0.902 | 24.21 | 0.897 |
| DSRNet† [23] | 23.91 | 0.818 | 26.74 | 0.920 | 24.83 | 0.911 | 26.11 | 0.906 | 25.75 | 0.910 |
| RRW†△ [65] | 21.83 | 0.801 | 26.67 | 0.931 | 24.04 | 0.903 | 26.49 | 0.915 | 25.34 | 0.912 |
| Zhong et al.†* [64] | 24.05 | 0.824 | 26.51 | 0.927 | 25.02 | 0.915 | 26.23 | **0.925** | 25.75 | 0.917 |
| Ours† | **25.22** | **0.836** | **27.27** | **0.932** | **25.58** | **0.922** | **27.40** | 0.918 | **26.49** | **0.922** |

Table 2: Quantitative results on the "Nature" testings set SIRS methods trained under data setting II. The best results are shown in **bold**, and the second-best are underlined.

| Metrics | ERRNet-F | IBCLN | YTMT | Dong et al. | DSRNet | RRW | Zhong et al. | Ours |
|---|---|---|---|---|---|---|---|---|
| PSNR | 22.18 | 23.57 | 23.85 | 23.45 | 25.22 | 26.04 | 23.87 | **26.77** |
| SSIM | 0.756 | 0.783 | 0.810 | 0.808 | 0.832 | 0.846 | 0.812 | **0.847** |

**Gradient reconstruction loss.** Considering the gradient independence, as a traditional prior in blind-source decomposition, we simultaneously encourage the models to restore the ground-truth gradient and penalize the intersected gradient as follows:

$$\mathcal{L}_{grad} := \|\nabla\hat{\mathbf{T}} - \nabla\mathbf{T}\|_1 + \|\nabla\hat{\mathbf{R}} - \nabla\mathbf{R}\|_1 + \frac{1}{N}\sum_{n=0}^{N-1}\beta\|\mathcal{D}(\hat{\mathbf{T}}^{\downarrow n}, \hat{\mathbf{R}}^{\downarrow n})\|_2^2,$$

$$\mathcal{D}(\hat{\mathbf{T}}, \hat{\mathbf{R}}) := \tanh\left(\xi_1|\nabla\hat{\mathbf{T}}|\right) \circ \tanh\left(\xi_2|\nabla\hat{\mathbf{R}}|\right),$$

(8)

where $\nabla$ denotes the difference operator of images. $\hat{\mathbf{T}}^{\downarrow n}, \hat{\mathbf{R}}^{\downarrow n}$ are $2^n$ down-sampled version of $\hat{\mathbf{T}}$ and $\hat{\mathbf{R}}$. $\xi_1$ and $\xi_2$ are normalization factors. The exclusion term, introduced by [60], ensures the multi-scale exclusion of the two layers in the gradient domain.

**Feature reconstruction Loss.** To promote the perceived quality of decoupled layers, we harness the following feature reconstruction loss:

$$\mathcal{L}_{fea} := \sum_i \omega_i \|\phi_i(\hat{\mathbf{T}}) - \phi_i(\mathbf{T})\|_1,$$

(9)

where $\phi_i(\cdot)$ represents the intermediate feature of the pre-trained VGG-19 model, where $i \in \{2, 7, 12, 21, 30\}$ tells the layer id. $\omega_i$ balance the weights of hierarchies.

**Total Loss.** The full training objectives $\mathcal{L}_{total}$ is defined as follows:

$$\mathcal{L}_{total} := \lambda_1 \mathcal{L}_{pix} + \lambda_2 \mathcal{L}_{grad} + \lambda_3 \mathcal{L}_{fea},$$

(10)

where $\lambda_1 = 1$, $\lambda_2 = 1$, and $\lambda_3 = 0.01$ are coefficients for balancing different loss terms.

## 4 Experimental Validation

### 4.1 Implementation Details

**Datasets**. Our training datasets include both synthetic and real-world images. Following [23], we design two data settings for fair comparison: I. 7,643 synthesized pairs randomly sampled from the

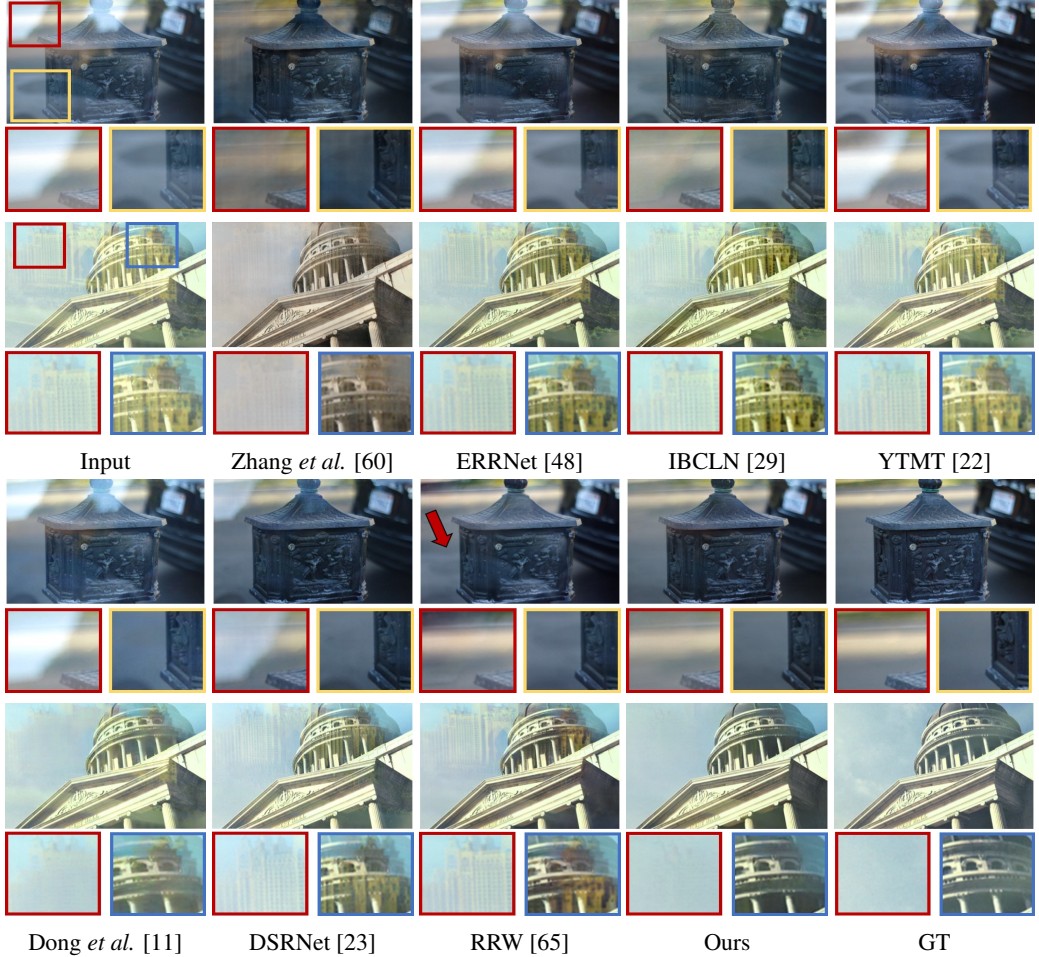

|  |  |  |  |  |
|---|---|---|---|---|
| Input | Zhang *et al.* [60] | ERRNet [48] | IBCLN [29] | YTMT [22] |
| Dong *et al.* [11] | DSRNet [23] | RRW [65] | Ours | GT |

Figure 4: Visual comparison of transmission layer predictions between previous state-of-the-arts and ours on samples from Real20 [60] and SIR$^2$ datasets. Please note the areas in the boxes.

PASCAL VOC dataset [13] in each epoch and 90 real pairs from [60]. II. 200 extra real pairs from the "Nature" dataset [29], and 13,700 synthesized pairs sampled from [60] instead. The training image size is fixed as $384 \times 384$. The window size of attention mechanisms, $N_W$, is fixed to $12 \times 12$, and the number of windows, $N_T$, varies depending on the spatial scale of the features.

**Training Strategy**. Our models are all implemented via the PyTorch framework and optimized with Adam optimizer for 20 or 80 epochs based on different data settings. The learning rate is fixed as $10^{-4}$ with a batch size of 1 on a single RTX 3090 GPU. Given real-world data pairs are hard to acquire, we additionally propose a data augmentation operation Reflection Mixup (RefMix) for the training real pairs, formulated as $\mathbf{I}_{\text{aug}} = \gamma\mathbf{I} + (1 - \gamma)\mathbf{T}$, where $\gamma \in [0, 1]$ is uniformly sampled.

### 4.2 Performance Evaluation

**Quantitative comparison**. As shown in Tabs. 1 and 2, we make a comparison between ours and state-of-the-art methods on five real-world datasets, including Real20 [60], Nature20 [29] and three subsets of the SIR$^2$ Dataset [42]. It is noteworthy that our models trained on both data settings show superior performance over the alternatives on most testing sets, including those that involve extra real-world data [65] and language prompts [64]. The superiority is attributed not only to the improved generalizability afforded by the hybrid Transformer architecture but also to the dual-attention interactive design that directly assesses intra-layer and inter-layer correlations, which shows impressive efficiency on SIRS tasks and has a high potential for other decomposition tasks.

**Qualitative comparison**. To evaluate our proposed model aesthetically, we first present a visual comparison of estimated transmission layers in Fig. 4. The two superimposed input images are

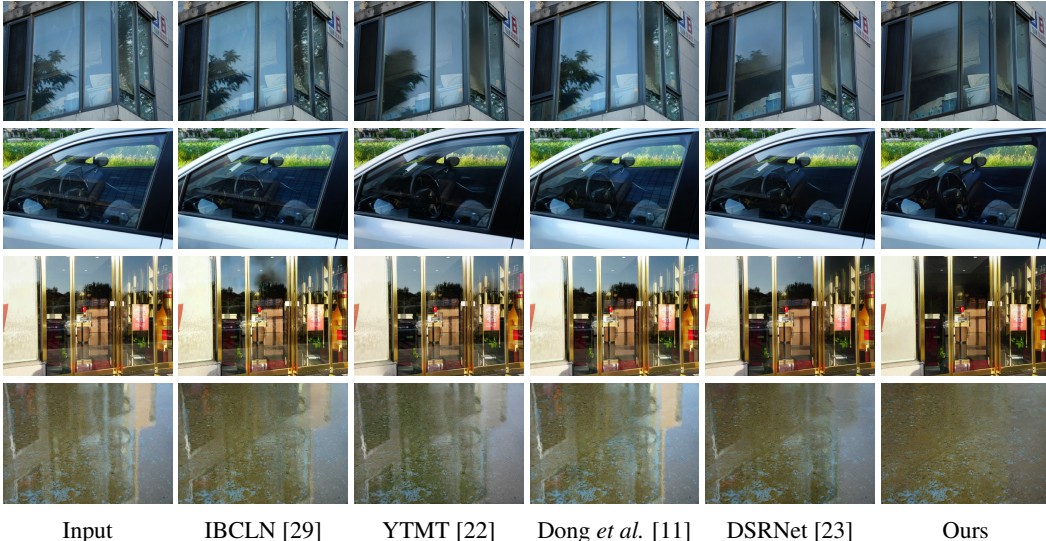

| Input | IBCLN [29] | YTMT [22] | Dong *et al.* [11] | DSRNet [23] | Ours |

Figure 5: Visual comparison of transmission predictions between previous state-of-the-arts and ours in real-world scenarios additionally captured in this paper. The broad advantages demonstrated by our method across these diverse conditions highlight its superior generalization capability.

sampled from Real20 and SIR$^2$ datasets, respectively. The two cases are representative since the first case is captured outdoors and contains both specular and weak reflections, while the second one is taken indoors with a relatively uniform reflection layer, which is highly entangled with the transmission structure. As can be seen, for the first case, ERRNet, YTMT, and Dong *et al.* cannot recognize the reflection regions successfully. Zhang *et al.* removes the reflections at the cost of introducing color bias and artifacts. IBCLN, DSRNet, and RRW separate either the specular or the weak reflection parts, lacking the ability to correlate the reflection components of different intensities. As for the second one, most alternatives fail to separate the majority of the reflection layer. Although Dong *et al.* shows an improvement over previous methods, it still leaves blurry reflection components in its result transmission layer. With a better layer modeling capability, our models conquer such a problem, providing strikingly clearer reconstructions.

Additionally, we specifically captured several in-the-wild test cases, as illustrated in Fig. 5. Unlike standard benchmarks that often incorporate artificial glass plates, these cases utilize real-world reflective surfaces, including reflections from accumulated water—conditions entirely absent from the training set. The superior performance of our approach, in comparison to previous state-of-the-art methods, underscores its robust generalization capacity and practical applicability. We further compare the reflection predictions between dual-stream reflection separation models in Fig. 6. Notably, our DSIT model yields significantly more plausible results, exhibiting superior content fidelity and color accuracy, attributed to the enhanced information selection capabilities of our design.

### 4.3 Ablation Study

**Model selection for GPE**. Generally speaking, the design of our DAIE allows different choices of global prior extractors, which aim to provide semantic priors and/or non-local information. To elaborate on the efficacy of different GPE models, we compare the settings of using ResNet101 [21], FocalNet-L [52], PVTv2-b4 [45], and Swin-L [35] models, which are all pre-trained on image classification tasks and finetuned on object detection tasks. As shown in Tab. 3, the ResNet101 as a CNN backbone provides limited global priors and poorly generalizes to the Real20 dataset. PVTv2 comprises few inductive biases, thus exhibiting inferior performance to the FocalNet and Swin Transformer, which preserve local dependency.

**Design of CAI operation**. Our cross-architecture interaction mechanism is designed to exploit useful information from the priors extracted by GPE and LPE, and thus, various feature fusion operations can be taken. Here, we build a baseline "Add" that simply sums the information flow provided by the different prior extractors. Further, we propose two variants based on Cross Attention design ("CrossAttn") and DAIB ("DAIB") mechanisms, respectively. As Tab. 3 shown, cross-attention is not

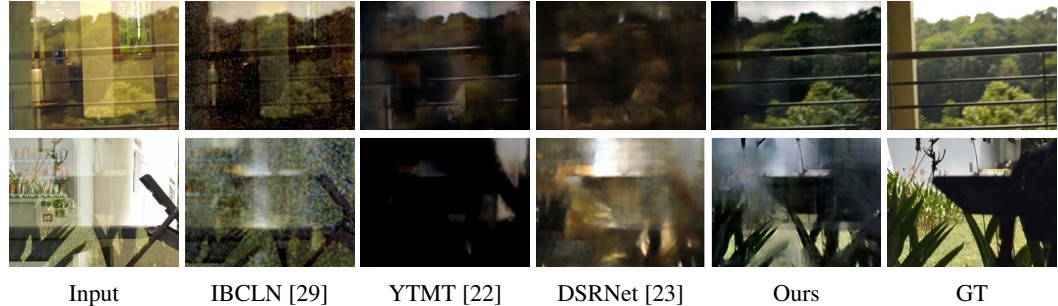

| Input | IBCLN [29] | YTMT [22] | DSRNet [23] | Ours | GT |

Figure 6: Visual comparison of reflection layer predictions between previous state-of-the-arts and ours on the SIR$^2$ dataset. Our method shows significant superiority over previous dual-stream arts.

capable of fusion cross-architecture information, leading to worse performance than simple addition, while our proposed DAIB bridges the local and global priors, obtaining superior results.

**Design of DAIB module**. To demonstrate the effectiveness of our DAIB design, we constructed three baseline models. The first baseline, "MLP FFN", replaces our DSLP Block with a standard MLP module. The second, "w/o DS-CA", omits the DS-CA mechanism, and the third, "w/o DS-SA", removes the DS-SA mechanism. As shown in Tab. 3, model performance degrades when substituting DSLP Block with MLP, primarily due to the reduced inductive biases and feature interactions. Moreover, the removal of either DS-CA or DS-SA results in inferior performance, particularly for DS-CA. This highlights the critical role of our proposed dual-attention interactive mechanism.

**Design of LRPB mechanism**. LRPB provides an initial bias to each attention

Table 3: Ablation study on different factors of our design.

| Factors | Instances | Real20 (20) | | SIR$^2$ (454) | |
|---|---|---|---|---|---|
| | | PSNR | SSIM | PSNR | SSIM |
| GPE | ResNet101 | 21.59 | 0.777 | 24.42 | 0.886 |
| | PVTv2-b4 | 23.16 | 0.793 | 24.44 | 0.890 |
| | FocalNet-L | 24.13 | 0.824 | 25.52 | 0.910 |
| | Swin-L | **25.06** | **0.836** | **26.32** | **0.920** |
| CAI | Add | 24.62 | 0.825 | 25.79 | 0.917 |
| | CrossAttn | 24.50 | 0.819 | 24.76 | 0.896 |
| | DAIB | **25.06** | **0.836** | **26.32** | **0.920** |
| DAIB | MLP FFN | 23.65 | 0.817 | 25.38 | 0.909 |
| | w/o DS-CA | 24.48 | 0.823 | 25.89 | 0.919 |
| | w/o DS-SA | 24.47 | 0.827 | 26.12 | 0.920 |
| | DAIB | **25.06** | **0.836** | **26.32** | **0.920** |
| LRPB | w/o LRPB | 24.93 | 0.821 | 25.99 | 0.917 |
| | w/ LRPB | **25.06** | **0.836** | **26.32** | **0.920** |
| RefMix | w/o RefMix | 24.72 | 0.823 | 26.27 | 0.915 |
| | w/ RefMix | **25.06** | **0.836** | **26.32** | **0.920** |

point according to the spatial and layer locations, injecting a layer-aware prior into the attention mechanism. Through the ablation study of whether or not to equip with LRPB shown in Tab. 3 "LRPB", we show its merit in handling the layer decomposition problem.

**The usage of RefMix**. To evaluate the effectiveness of our proposed RefMix, a comparison of employing it or not is made in Tab. 3 "RefMix", showing that it aids the reflection separation.

Due to the page limitation, more visual analyses are organized in the appendix.

## 5 Concluding Remarks

In this study, a dual-stream interactive Transformer has been designed to address the challenge of single image reflection separation. To harness high-quality priors from pre-trained Transformer models, we developed a dual-architecture interactive encoder, which can effectively fuse multi-source information with adaptive emphases. Additionally, we introduced a novel dual-attention interactive block that utilizes both an effective dual-stream self-attention mechanism and a layer-aware dual-stream cross-attention module to separate the entangled features. Comprehensive experiments together with ablations have been conducted to verify the advances of our method. Looking forward, several interesting points deserve future exploration. For instance, larger vision foundation models can provide more substantial priors for low-level vision tasks. A relative position bias design tailored for a specific vision task will be also beneficial. Furthermore, our RefMix technique is likely to extend its benefits to other data-hungry visual tasks. These considerations may inspire future research on low-level Transformer designs.

## Acknowledgement

This work was supported by the National Natural Science Foundation of China under Grant nos. 62372251 and 62072327.

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

# A    Appendix

## A.1    Algorithm for Dual-Attention Interactive Block

We offer the overall procedure of the DAIB in Alg. 1, corresponding to the description in Sec. 3.1.

## A.2    Visual Analysis of Ablation Study

As shown in Fig. 7, we provide visual results of our ablation study. Among them, "GPE-ResNet101", "GPE-PVTv2-b4", and "GPE-FocalNet-L" show the effects of selecting different GPE models. ResNet101 provides limited global priors, thus exhibiting trivial performance. DSIT with PVTv2-b4 tends to concentrate on the wrong components during the layer decomposition due to a lack of inductive biases, producing unpleasant results. FocalNet-L replaces explicit query-key-value interactions with a focal modulation module. Although it shows merits over the former two settings, the lack of explicit information assessment limits its usability in our framework. The settings of "CAI-Add" and "CAI-CrossAttn" fail to effectively integrate global and local priors, resulting in inferior performance and misguided attention. "DAIB-MLP FFN" replaces the DSLP Block with MLP FFN, which also cannot separate the reflection successfully due to insufficient inductive biases. Moreover, we can observe that removing either attention mechanism in our dual-attention interaction design leads to appreciable performance degradation. Layered Relative Position Bias (LRPB) provides layer-aware attention initialization, while "w/p LRPB" loses the ability to separate reflections in some areas. As shown by "w/o RefMix", specific reflection patterns can be hard to separate after eliminating the reflection mixup augmentation.

## A.3    Visual Illustration of the RefMix

We visualize the results of our Reflection Mixup (RefMix) augmentation in Fig. 8. The hyperparameter $\gamma$ controls the blending rate of $\mathbf{I}$ and $\mathbf{T}$. Since the image pairs are aligned, adjusting their blending rate is actually tuning the intensity of the reflection layer. Some regions covered by strong reflections can thereby be learned reasonably.

## A.4    More Visual Comparisons

We provide more visual comparisons of estimated transmission and reflection layers in Figs. 9-12. The overall superiority shown by our models demonstrates the effectiveness of our DSIT design.

Note that the case is sampled from the Real20 dataset in Fig. 12, and no ground-truth reflection layer is provided with the dataset, so we put the result of the input image subtracted by the ground-truth transmission layer as the reference. It is clear that the hollows should not appear in the real reflection scene. Our models present a potential ability of inpainting these areas with the aid of global priors.

## A.5    Limitations

Previous SIRS methods typically fail to deal with the regions dominated by reflections, which show negligible defocus effects, as shown in Fig. 13. Although our model takes a significant step forward to solve the hard case beyond the former methods, it still leaves some visible reflections due to the deviation of the case from the relative smoothness assumption. We can expect that it will be better handled when involving more real-world pairs.

**Algorithm 1** The computational process for Dual-Attention Interactive Block.

---

**Require:** Dual-stream features $\mathbf{F}_{\mathbf{T}}^{\text{IN}}$ and $\mathbf{F}_{\mathbf{R}}^{\text{IN}}$.

**Ensure:** Interacted dual-stream features $\mathbf{F}_{\mathbf{T}}^{\text{OT}}$ and $\mathbf{F}_{\mathbf{R}}^{\text{OT}}$.

1: LayerNorm:
2:      $\mathbf{F}_{\mathbf{T}}^{\text{LN}} = \text{LN}(\mathbf{F}_{\mathbf{T}}^{\text{IN}}), \mathbf{F}_{\mathbf{R}}^{\text{LN}} = \text{LN}(\mathbf{F}_{\mathbf{R}}^{\text{IN}})$
3: Combine tokens both inter/intra-window-wise:
4:      $\mathbf{X}_0^{\text{LN}} = \text{Concat}([\mathbf{F}_{\mathbf{T}}^{\text{LN}}, \mathbf{F}_{\mathbf{R}}^{\text{LN}}], \text{dim}=0)$
5:      $\mathbf{X}_1^{\text{LN}} = \text{Concat}([\mathbf{F}_{\mathbf{T}}^{\text{LN}}, \mathbf{F}_{\mathbf{R}}^{\text{LN}}], \text{dim}=1)$
6: Apply DS-SA and DS-CA, respectively:
7:      $\mathbf{X}_0^{\text{SA}} = \text{DS\_SA}(\mathbf{X}_0^{\text{LN}})$
8:      $\mathbf{X}_1^{\text{CA}} = \text{DS\_CA}(\mathbf{X}_1^{\text{LN}})$
9: Split tokens back to dual streams:
10:      $\mathbf{F}_{\mathbf{T}}^{\text{SA}}, \mathbf{F}_{\mathbf{R}}^{\text{SA}} = \text{Split}(\mathbf{X}_0^{\text{SA}}, \text{dim}=0)$
11:      $\mathbf{F}_{\mathbf{T}}^{\text{CA}}, \mathbf{F}_{\mathbf{R}}^{\text{CA}} = \text{Split}(\mathbf{X}_1^{\text{CA}}, \text{dim}=1)$
12: Combine the dual-attention results with skip connections:
13:      $\mathbf{F}_{\mathbf{T}}^{\text{DA}} = \mathbf{F}_{\mathbf{T}}^{\text{IN}} + \mathbf{F}_{\mathbf{T}}^{\text{SA}} + \mathbf{F}_{\mathbf{T}}^{\text{CA}}$
14:      $\mathbf{F}_{\mathbf{R}}^{\text{DA}} = \mathbf{F}_{\mathbf{R}}^{\text{IN}} + \mathbf{F}_{\mathbf{R}}^{\text{SA}} + \mathbf{F}_{\mathbf{R}}^{\text{CA}}$
15: Apply the DSLP Block as FFN:
16:      $\mathbf{F}_{\mathbf{T}}^{\text{FFN}}, \mathbf{F}_{\mathbf{R}}^{\text{FFN}} = \text{DSLP\_Block}(\text{LN}(\mathbf{F}_{\mathbf{T}}^{\text{DA}}), \text{LN}(\mathbf{F}_{\mathbf{R}}^{\text{DA}}))$
17: Derive the outputs $\mathbf{F}_{\mathbf{T}}^{\text{OT}}, \mathbf{F}_{\mathbf{R}}^{\text{OT}}$:
18:      $\mathbf{F}_{\mathbf{T}}^{\text{OT}} = \mathbf{F}_{\mathbf{T}}^{\text{DA}} + \mathbf{F}_{\mathbf{T}}^{\text{FFN}}$
19:      $\mathbf{F}_{\mathbf{R}}^{\text{OT}} = \mathbf{F}_{\mathbf{R}}^{\text{DA}} + \mathbf{F}_{\mathbf{R}}^{\text{FFN}}$

---

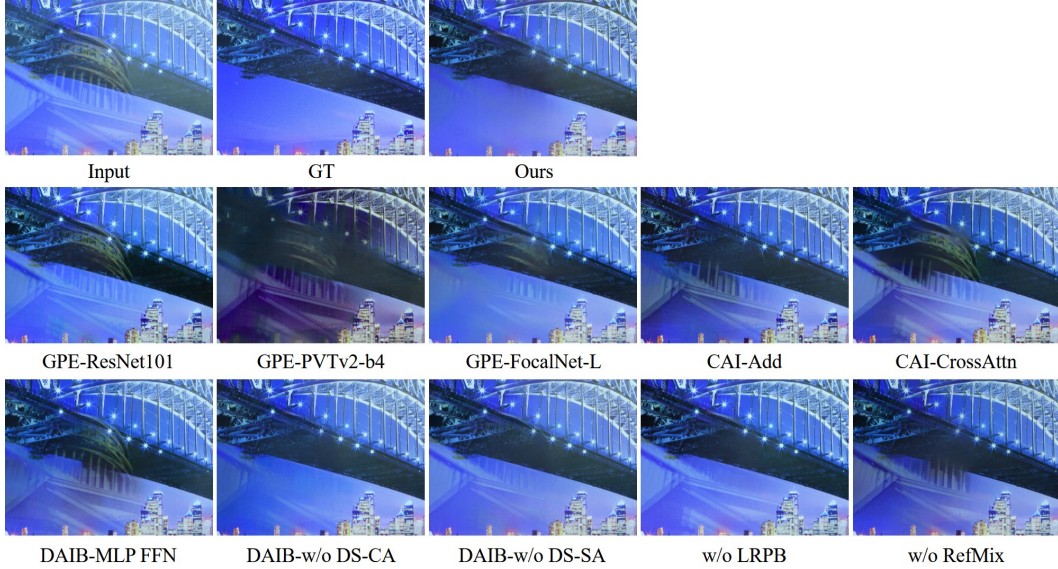

Figure 7: Visual results of DSIT variants shown in Tab. 3. The case is sampled from the SIR$^2$ dataset.

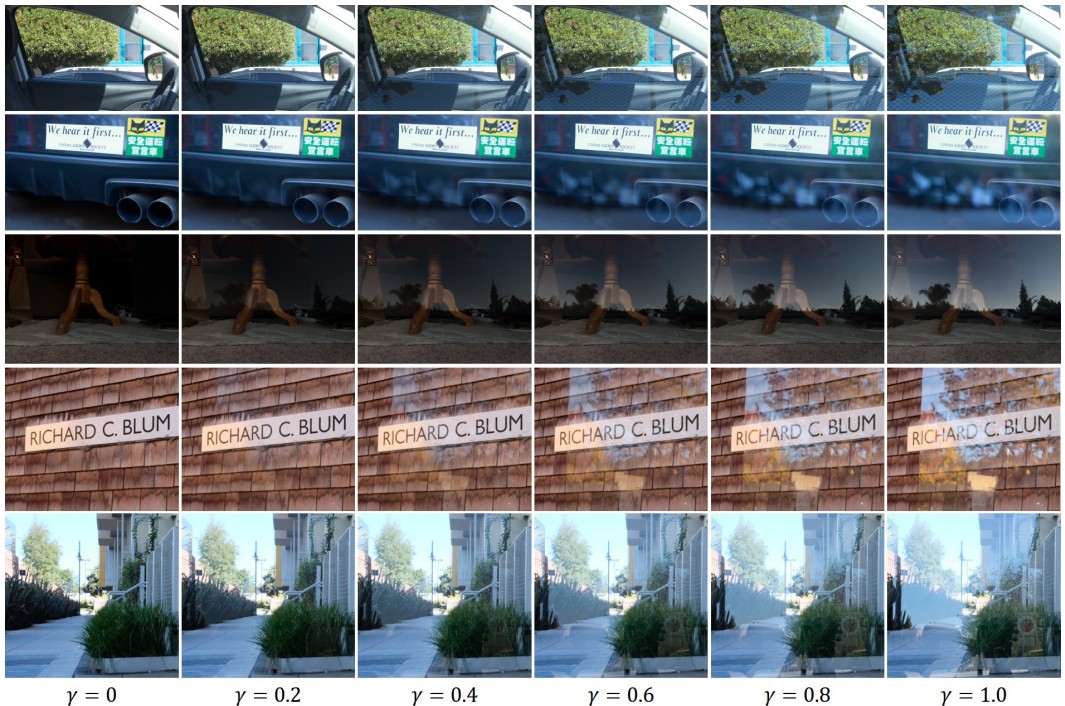

| $\gamma = 0$ | $\gamma = 0.2$ | $\gamma = 0.4$ | $\gamma = 0.6$ | $\gamma = 0.8$ | $\gamma = 1.0$ |

Figure 8: The RefMix results according to different $\gamma$s, which enrich the real pairs during training.

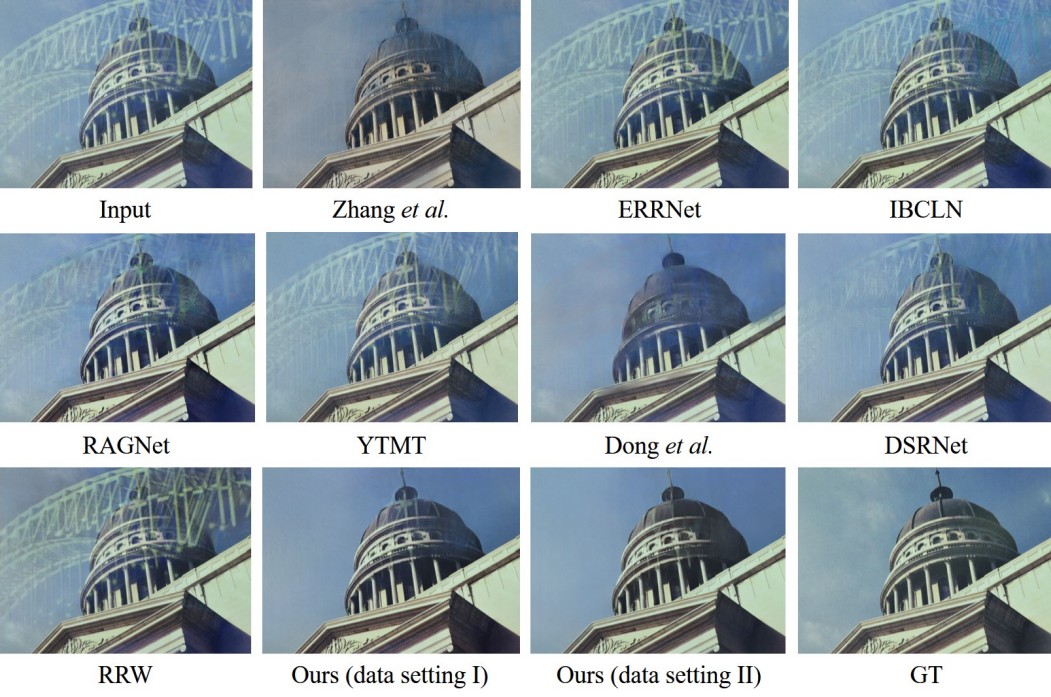

| Input | Zhang *et al.* | ERRNet | IBCLN |
| RAGNet | YTMT | Dong *et al.* | DSRNet |
| RRW | Ours (data setting I) | Ours (data setting II) | GT |

Figure 9: Visual comparison of transmission layer predictions between previous arts and ours. The case is sampled from the SIR$^2$ dataset.

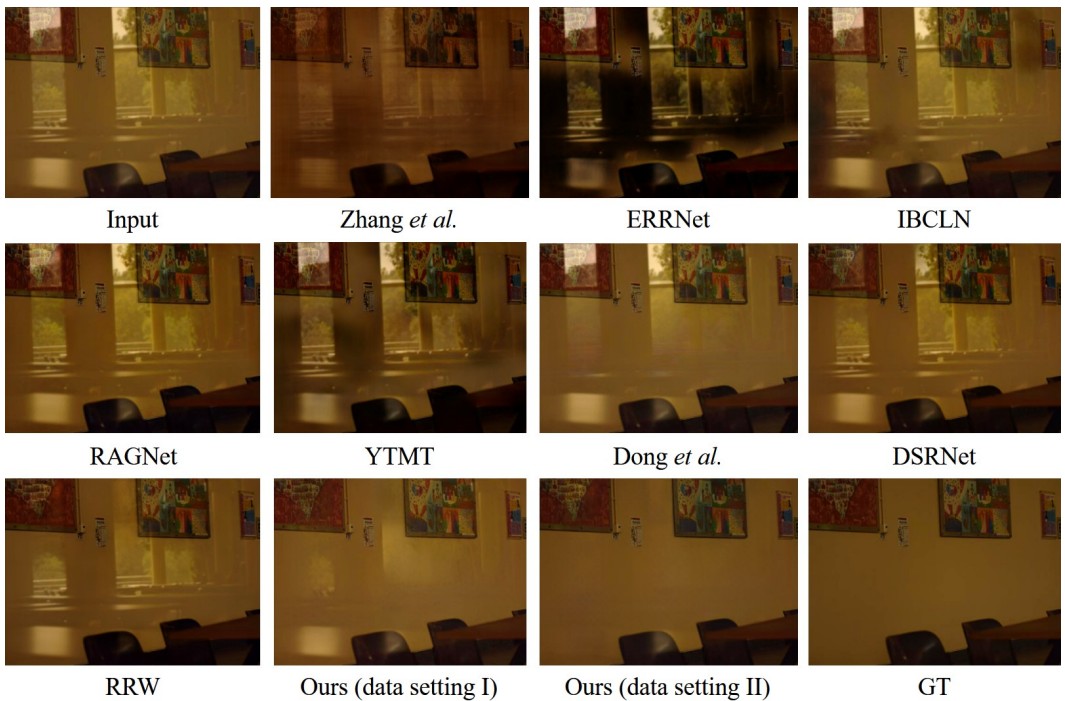

Figure 10: Visual comparison of transmission layer predictions between previous arts and ours. The case is sampled from the SIR$^2$ dataset.

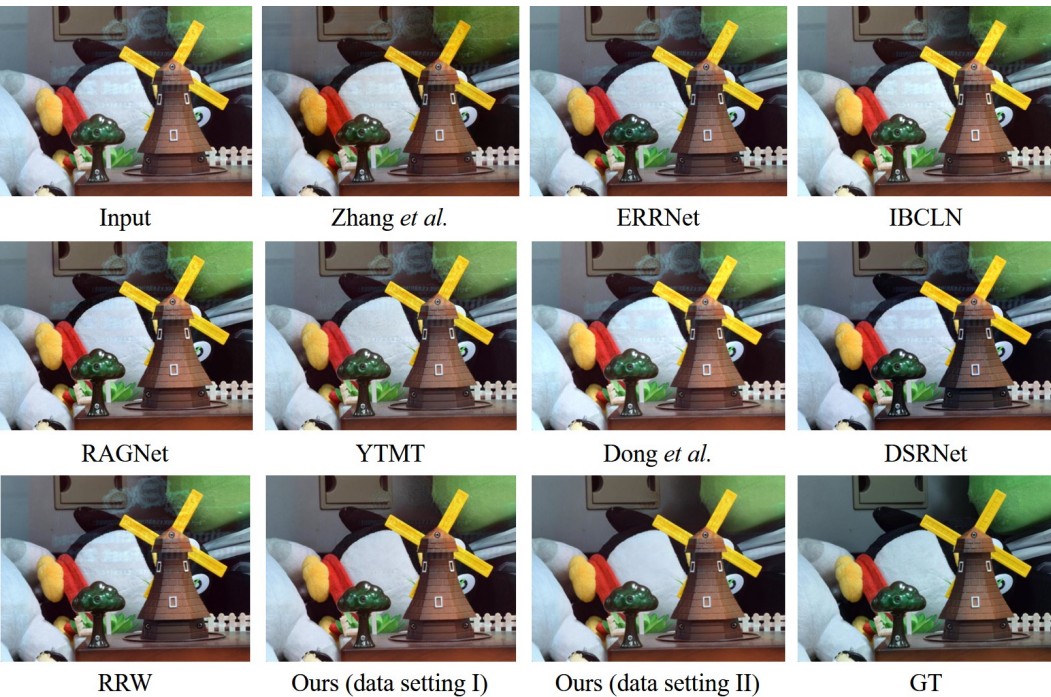

Figure 11: Visual comparison of transmission layer predictions between previous arts and ours. The case is sampled from the SIR$^2$ dataset.

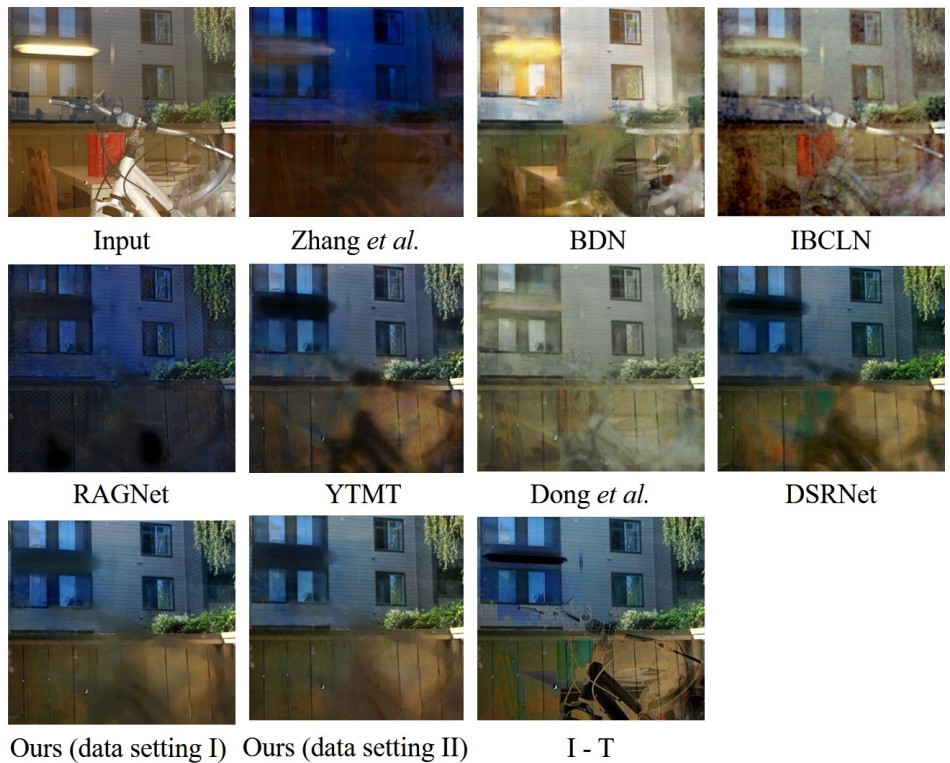

Figure 12: Visual comparison of reflection layer predictions between previous arts and ours.

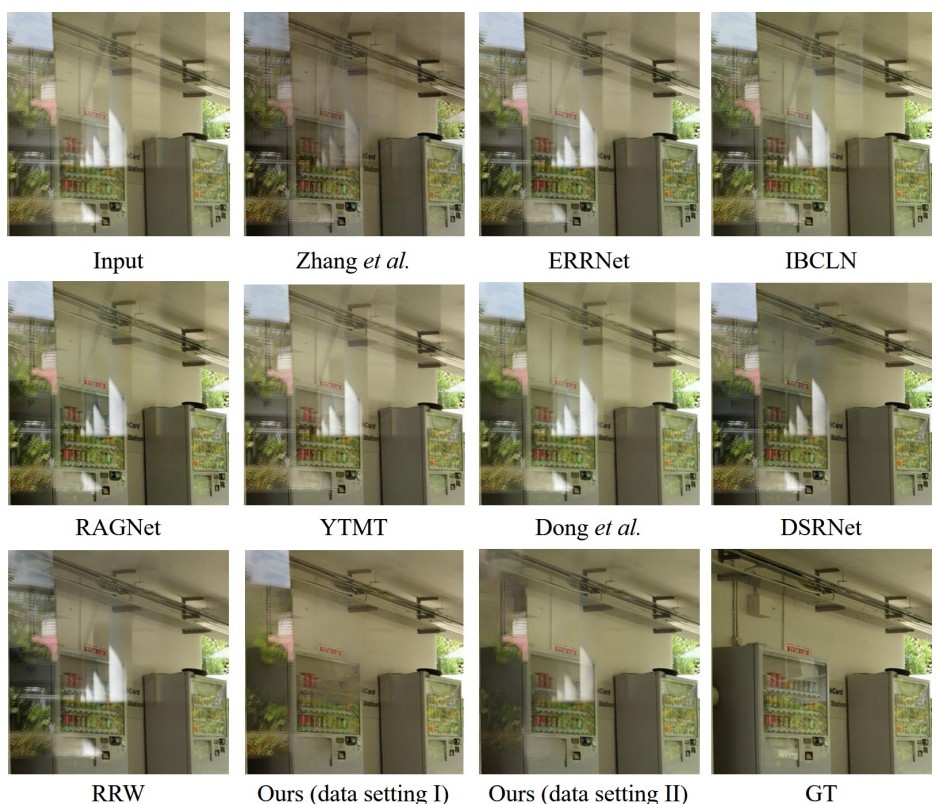

Figure 13: Our limitation illustration, when solving a hard case, sampled from the SIR$^2$ dataset.

