# OpenReview forum: "Single Image Reflection Separation via Dual-Stream Interactive Transformers"
_NeurIPS.cc/2024/Conference — NeurIPS 2024 poster_

### Official Review · Reviewer_gpBZ · 2024-07-08

**Soundness:** 3
**Presentation:** 3
**Contribution:** 2
**Rating:** 5
**Confidence:** 3

**Summary:**

This work introduces ADI, a new interactive dual-stream transformer framework for single image reflection separation. It incorporates a dual-attention interaction to explicitly model the dual-stream correlation for improved reflection separation, achieving impressive performance compared to other SOTA methods.

**Strengths:**

1. This paper addresses the limited interaction of previous dual-stream transformer frameworks for the SIRS task by presenting the ADI module, which simultaneously considers inter-window and intra-window fusion.
2. The quantitative results are impressive compared to previous SOTA works.
3. The discussion and motivation are well-explained and intuitively correct.

**Weaknesses:**

1. The design is quite plain, involving inter-patch/intra-patch attention.
2. The choice of using gradient and feature loss should be analyzed in ablation studies to verify their contributions.
3. Model complexity and inference time are missing in the main manuscript, making it difficult to determine whether the main contribution comes from the larger model capacity or the proposed algorithm.

**Questions:**

see the weaknesses.

---

> ### Author Rebuttal · Authors · 2024-08-07
>
> Thank you for recognizing the motivation, technical soundness, and state-of-the-art performance of our method. Below, we address the key concerns you raised:
>
> **Q1**: Why the design of inter-patch/intra-patch attention is quite plain?
>
> **A1**: Keeping the design simple is beneficial to validate its effectiveness. When the pipeline becomes complicated, it can be hard to determine which part of the design really works. Also, more powerful and complicated attention designs can be developed in future work, which can potentially enhance the performance further.
>
> Through the derivation of LaCA (Layer-aware Cross-Attention) provided in the global Author Rebuttal, it is evident that even with a straightforward token concatenation operation and corresponding modifications to the relative position bias, we naturally introduce inter- and intra-layer attention for dual-stream information. This demonstrates the inherent relationship between dot-product self-attention mechanisms and dual-stream interactive modeling.
>
> In the meantime, our strategy can also energize other multi-stream transformer designs. For instance, in high-resolution image generation scenarios where a single GPU cannot support the entire image generation process, multiple information streams of the model can be distributed across different GPUs. Each stream would handle a specific image patch, and inter-stream feature interactions would ensure the coherence of the generated patches, enabling them to be seamlessly stitched together into a complete image (like in [1]).
>
> **Q2**: What about the contributions of gradient and feature losses?
>
> **A2**: Both gradient and feature losses contribute positively to our method. The importance of gradient relationship has been emphasized by early works in SIRS [2],[3],[4]. Feature (or perceptual) loss, on the other hand, significantly aids in modeling natural images by ensuring that the reconstructed images are perceptually similar to the ground truth [5],[6].
> As suggested by the reviewer, to confirm their contributions, we additionally conducted ablation experiments, as shown in the table below. We evaluated the impact of removing Gradient Loss (DSIT (w/o Gradient Loss)) and Feature Loss (DSIT (w/o Feature Loss)) on the performance of our model. The results indicate a noticeable decrease in performance when these losses are disabled, underscoring their necessity. Specifically, the gradient loss helps preserve edge details, while the feature loss enhances the perceptual quality of the outputs, both of which are beneficial for effective reflection separation.
>
> |          Models         |      Real20     |     Objects     |     Postcard    |       Wild      | Average PSNR/SSIM |
> |:-----------------------:|:---------------:|:---------------:|:---------------:|:---------------:|:-----------------:|
> | DSIT (w/o Gradient Loss) |   24.13/0.814   |   27.54/0.922   |   23.38/0.898   |   26.72/0.906   |    25.55/0.906    |
> |  DSIT (w/o Feature Loss) |   23.13/0.780   |   26.00/0.919   |   23.80/0.896   |   25.88/0.900   |    24.94/0.901    |
> |        DSIT(Ours)       | **25.06/0.836** | **26.81/0.919** | **25.63/0.924** | **27.06/0.910** |  **26.27/0.917**  |
> |
>
> **Q3**: Please provide the model complexity and inference time.
>
> **A3**: We have provided a comparison in model parameters, GFLOPs (for $384 \times 384$ resolution inputs), inference time (for $384 \times 384$ resolution inputs, tested on an RTX 3090 GPU), and the average PSNR/SSIM across 4 datasets (Real20, Objects, Postcards, and Wild) in the table below. The number of parameters of our model is comparable to RAGNet and DSRNet. Due to our multi-scale and non-recursive network structure, we achieve superior GFLOPs efficiency compared to all other methods. Despite being based on a Transformer architecture, our method demonstrates a slightly faster inference time than DSRNet, highlighting the efficiency of our design. The above results show that the improvement of our model is non-trivial. As further illustrated in the additional cases provided in the global rebuttal PDF, our method exhibits impressive generalization performance in real-world scenarios compared to previous state-of-the-art approaches. This reflects the strong capability in effectively modeling reflection scenes of our method.
>
> |        Models       |    ERRNet   |    IBCLN    |    RAGNet   |     YTMT    | Dong _et al._ |    DSRNet   |       Ours      |
> |:-------------------:|:-----------:|:-----------:|:-----------:|:-----------:|:-------------:|:-----------:|:---------------:|
> |    Parameters (M)    |    18.58    |    24.37    |    146.72   |    76.90    |   **10.93**   |    124.60   |      131.76     |
> |        GFLOPs       |     820     |     708     |     610     |     838     |      666      |     743     |     **517**     |
> | Inference Time (ms) |      72     |      75     |    **56**   |     101     |      123      |     256     |       215       |
> | Average PSNR/SSIM| 23.53/0.879 | 24.10/0.879 | 24.90/0.886 | 24.05/0.886 |  24.21/0.897  | 25.75/0.910 | **26.49/0.922** |
> |
>
> [1] DistriFusion: Distributed Parallel Inference for High-Resolution Diffusion Models. CVPR 2024.
>
> [2] User Assisted Separation of Reflections from a Single Image Using a Sparsity Prior. ECCV 2004.
>
> [3] Single Image Layer Separation Using Relative Smoothness. CVPR 2014.
>
> [4] A Generic Deep Architecture for Single Image Reflection Removal and Image Smoothing. ICCV 2017.
>
> [5] Image style transfer using convolutional neural networks. CVPR 2016.
>
> [6] Perceptual losses for real-time style transfer and super-resolution. ECCV 2016.

---

> > ### Comment · Reviewer_gpBZ · 2024-08-13
> > **rating**
> >
> > Thanks for your response. I'd like to keep my rating as 'Borderline accept'.

---

> > > ### Author Response · Authors · 2024-08-14
> > >
> > > Thank you for your efforts in reviewing our manuscript. If you have any further concerns, we welcome continued discussion and would be happy to address them.

---

### Official Review · Reviewer_9gCf · 2024-07-10

**Soundness:** 3
**Presentation:** 2
**Contribution:** 3
**Rating:** 5
**Confidence:** 4

**Summary:**

The paper proposes a transformer-based network for the single image reflection separation task. It focuses on the network architecture design, proposing several tailored designs, including dual-attention interactive block (DAIB), dual-stream self-attention (DS-SA), and dual-stream cross-attention (DS-CA).

**Strengths:**

1.	The introduction is well-written.
2.	The proposed method demonstrates state-of-the-art (SOTA) performance on the traditional single image reflection separation datasets.

**Weaknesses:**

1.	The method section is poorly written, making it difficult to follow due to the extensive use of abbreviations, some of which are not explained upon first mention, such as DS-SA and DS-CA.
2.	Despite a thorough reading of the methods and implementation details, it is unclear where the authors have applied the pretrained transformer. Specifically, I mean "pretrained transformer." Additionally, the placement and role of the proposed CNN-based network within the DSIT framework are not clearly clarified.
3.	For Eq. (3), the authors should clarify where the 'cross' is present. The equation seems to suggest a self-attention mechanism, as the query, key, and value all derive from the same features.
4.	In Fig. 3, it is unclear if the increased intensity of red indicates regions of higher attention. If so, I cannot distinguish the differences between the local and global priors as revealed by the figure.
5.	I think the current setting of single image reflection separation does not align well with typical real-world scenarios. Reflections in real scenes typically occur in localized regions, such as the reflection of a building with a glass facade. Therefore, relying heavily on synthetic reflections with minimal real-scene data might not be practical. The authors should consider focusing on real-scene reflections.
Overall, I think the novelty of the paper is limited, and the writing is poor.

**Questions:**

Please refer to the weaknesses for details.

**Limitations:**

Given the advancements in deep learning for computer vision, the proposed reflection separation task focuses on synthetic pairs. I suggest the authors shift their research focus towards reflections occurring in real-world scenes.

---

> ### Author Rebuttal · Authors · 2024-08-07
>
> Thank you for your detailed feedback. We appreciate your comments and try our best to address the issues as follows:
>
> **Q1**: The writing of the method section.
>
> **A1**: We apologize for any difficulties caused by the writing in the method section. In the next version, we will thoroughly proofread and clarify our paper. Specifically, we will ensure that the terms DS-SA and DS-CA are properly defined before their first use, as follows:
>
> "Following the LayerNorm, we apply Dual-Stream Self-Attention (DS-SA) to $\textbf{X}^{\text{LN}}\_0$ and Dual-Stream Cross-Attention (DS-CA) to $\textbf{X}^{\text{LN}}\_1$, obtaining $\textbf{X}\_{\text{SA}}$ and $\textbf{X}\_{\text{CA}}$, respectively. The design of these two attention mechanisms will be detailed later in this section."
>
> **Q2**: (1) The appliance of the pre-trained transformer. (2) The CNN-based network placement.
>
> **A2**: (1) We utilize a pretrained Transformer architecture for the Global Prior Extractor, as shown in Figure 2 (a) in the main paper. The transformer blocks in GPE can load pretrained weights before training, which can leverage rich priors learned from large-scale datasets to aid in reflection separation. This practice is consistent with previous reflection separation methods, which have similarly used pretrained models to enhance feature extraction, such as the HyperColumn used in Zhang et al. [1] and YTMT [2], and the DSFNet in DSRNet [3].
>
> (2) As depicted in Figure 2 (a), the Local Prior Extractor, referred to as the CNN-based Network, is implemented using a convolutional dual-stream interactive network structure. In our implementation, each DSLP Block specifically utilizes the MuGI Block from the DSRNet [3] to highlight our main contribution and avoid other possible factors influencing performance.  Additionally, more advanced convolutional modules can be developed in future work.
>
> We will highlight these points in our revision to avoid misunderstanding.
>
> **Q3**: Where is the "cross" presented?
>
> **A3**: We explain in detail how explicit correlation assessment works and where is the "cross" presented  in the A1 of common issues in the global rebuttal. We will clarify this point in our revised version.
>
> **Q4**: Elaboration on Figure 3.
>
> **A4**: In the figure, red regions in the feature visualization indicate high attention values, while blue regions represent low attention values. We will add a color bar in the revised version to clarify this.
>
> Please note that the pretrained model used in transfer learning is task-agnostic. We do not expect the priors alone to distinguish between the reflection and transmission layers clearly. Instead, the priors provide varying degrees of attention to different components of the image. For instance, the global prior $\mathbf{F}^{\text{GP}}$, extracted using a pretrained Swin Transformer model, shows higher attention towards reflection components. The shifted window-based self-attention contributes to the coherent attention across the image. Additionally, the Local Prior Extractor captures local prior information for both the transmission stream $\mathbf{F}^{\text{LP}}\_\mathbf{T}$ and the reflection stream $\mathbf{F}^{\text{LP}}\_\mathbf{R}$. These local priors exhibit a "granular" texture and may not accurately attend to the correct components of transmission and reflection. After the Cross-Architecture Interaction with the global prior $\mathbf{F}^{\text{GP}}$, the attention of $\mathbf{F}^{\text{LP}}\_\mathbf{T}$ and $\mathbf{F}^{\text{LP}}\_\mathbf{R}$ towards the transmission and reflection layers becomes more appropriately focused. Through further interaction and optimization of Dual-Attention Interactive Blocks, the dual-stream features $\mathbf{F}^{\text{DAIB}}\_\mathbf{T}$ and $\mathbf{F}^{\text{DAIB}}\_\mathbf{R}$ turn out to be more fine-grained and accurate. This improvement has also been recognized by Reviewer k1uS (Strengths 3).
>
> **Q5**: Does the current setting align with real-world scenarios?
>
> **A5**: Please note that all the testing data used in our paper are actually captured in the real world, including Real20, Nature20, Wild, Objects, and Postcard.  These datasets encompass a variety of indoor and outdoor scenes, different glass thicknesses, and varying distances, thereby covering a wide range of real-world reflection scenarios. Again, we emphasize that our method does *NOT* focus on synthetic scenarios, as the reviewer criticized.
>
> Meanwhile, during the rebuttal period, we captured several reflection scenes in the wild according to what the reviewer described as typical real-world scenarios. We compared the inference results of state-of-the-art methods and our approach in the global rebuttal PDF. In Figure 1 of the rebuttal PDF, the first 3 examples primarily contain localized reflection regions. The 4-th example showcases the reflection phenomenon on a water surface, which is not covered in the "minimal real-scene data" during training mentioned by the reviewer. Our results are visually impressive, demonstrating our model's generalization capability. Besides, Figure 2 of the rebuttal PDF presents the predicted reflection layers from our method on these real-world images, along with the corresponding binarized maps. From these visualizations, it is evident that our method can effectively identify localized reflection regions.
>
> Inspired by the suggestion of the reviewer, in future work, we may explore the introduction of a lightweight "reflection localizer". This would allow us to bypass the processing of non-reflective regions (like the manner of early exiting [4]), thereby accelerating the reflection separation process.
>
> [1] Single Image Reflection Separation with Perceptual Losses. CVPR 2018.
>
> [2] Trash or Treasure? An Interactive Dual-Stream Strategy for Single Image Reflection Separation.  NeurIPS 2021.
>
> [3] Single Image Reflection Separation via Component Synergy. ICCV 2023.
>
> [4] Adaptive Patch Exiting for Scalable Single Image Super-Resolution. ECCV 2022.

---

> > ### Comment · Reviewer_9gCf · 2024-08-09
> >
> > I appreciate the authors' efforts in their rebuttal, providing clearer explanations of the details and additional experimental results. These revisions have significantly addressed my major concerns regarding the novelty and contribution of the work, as well as the experimental design.
> >
> > Furthermore, the authors' commitment to refining the writing, particularly in Section 3, in the final version is commendable. Given these improvements, I feel positively about the manuscript and have decided to raise my final score to 5: Borderline Accept.

---

> > > ### Author Response · Authors · 2024-08-09
> > >
> > > We are glad that our responses have helped you gain a better understanding of our work. We are committed to improving the clarity and readability of our manuscript in the revised version. Thank you for your professional review and timely feedback. The constructive comments have significantly contributed to polishing our paper.

---

### Official Review · Reviewer_k1uS · 2024-07-11

**Soundness:** 3
**Presentation:** 2
**Contribution:** 3
**Rating:** 6
**Confidence:** 4

**Summary:**

The authors propose the Dual-Stream Interactive Transformer (DSIT) for single image reflection separation. DSIT is a dual-stream method designed for complex scenarios. Specifically, the authors design the Dual-Attention Interaction (DAI) to achieve feature aggregation of different dimensions through dual-stream self-attention and layer-aware cross-attention. They also propose the Dual-Architecture Interactive Encoder (DAIE) to leverage pre-trained Transformer models. Experiments demonstrate that the proposed method outperforms comparative methods.

**Strengths:**

1. The authors use DAI to enhance feature aggregation of transmission and reflection information, which is a reasonable approach. The ablation study also proves its effectiveness.
2. The authors introduce a pre-trained Transformer to provide global information and improve feature modeling. As shown in Figure 3, the final F_T^{DAIB} and F_R^{DAIB} have clearer detailed features.
3. Testing on multiple datasets shows that the proposed method outperforms state-of-the-art methods.
4. The authors conduct thorough experiments on various modules in the ablation study, further proving the effectiveness of the proposed method.

**Weaknesses:**

1. The implementation of CAI is not clearly explained. In L194, CAI seems to be through DAIB(F^{GP}, F_T^{LP}) and DAIB(F^{GP}, F_R^{LP}), while the proposed DAIB in the method deals with F_T and F_R. It is recommended that this be clarified in Figure 2 and the text to improve clarity. Additionally, the implementation of CAI is overly complex. Considering the high computational complexity of DAIB, a simpler implementation should be considered, such as directly using F GP in DAIB(F_T^{LP}, F_R^{LP}).
2. The authors do not provide detailed model settings, such as N_T, N_W, and the detailed structure of DSLP.
3. There is a lack of comparison between FLOPs and Params. The proposed method uses multiple attentions to aggregate features, which introduces high computational complexity.
4. In Figure 2, some parts of the Dual-Stream Information Flow overlap excessively, reducing distinguishability.

**Questions:**

1. Clarify the implementation of CAI and other model settings, such as N_T, N_W, and the structure of DSLP.
2. Provide a comparison of FLOPs in Tables 2 and 3.
3. The proposed DSIT uses a Transformer as a pre-trained model and claims to introduce prior knowledge. How does the model perform without using a pre-trained model? Additionally, why does DSLP not use pre-trained models?

**Limitations:**

The authors discuss the method's limitations and societal impacts.

---

> ### Author Rebuttal · Authors · 2024-08-07
>
> Thank you for your positive feedback on our work. Below, we address your main concerns:
>
> Q1: (1) Clarify the implementation of CAI, (2) N_T, N_W, and (3) DSLP.
>
> A1: (1) CAI is a general design introduced to enable interaction and fusion of features extracted from different network architectures during the feature extraction phase. A specific implementation of this design is the dual-stream feature interaction module. We tested 3 different implementations of CAI as in the ablation study shown in Table 3 of the main paper. It turned out that our current choice (DAIB) is significantly superior to other alternatives, which exhibits the effectiveness of DAIB for both cross-architecture and cross-layer interactions.
>
> (2) In our experiments, the training image size is fixed at $384\times384$. The window size of attention mechanisms, $N_W$, is fixed to $12\times12$. The number of windows, $N_T$, varies depending on the spatial scale of the features. For instance, if the spatial scale is $384\times384$, then $N_T = (384/12)\times(384/12)=1024$.
>
> (3) The DSLP can be any convolutional dual-stream network module, such as the YTMT Block [1] and MuGI Block [2]. To highlight our main contribution and avoid other possible factors influencing performance, we simply adopt the MuGI Block in our experiments. Also, more powerful convolutional modules can be developed in future work.
>
> We will clarify these model settings in our revised version and thoroughly proofread our paper to eliminate other writing problems.
>
> Q2: Simpler Implementation of DAIB.
>
> A2: In Table 3 of the main paper, we provided an ablation study on CAI. The "Add" variant represents the removal of the DAIB, directly adding the global priors on the dual-stream local priors. Although this setting reduces the complexity of the model (from 517 GFLOPs to 471 GFLOPs), it decreases the average PSNR from 26.27 to 25.74. To ensure superior performance, we retained the DAIB as the interaction module for CAI. However, considering broader applications, exploring more lightweight model designs can be more attractive, which may use a more lightweight backbone (MobileViT for example), reduce the number of channels and blocks, and employ knowledge distillation techniques. Thank you for your valuable suggestion, which inspires us to optimize the model for more practical use in the future.
>
> Q3: Provide a comparison of FLOPs and Params.
>
> A3: As shown in the Table below, we present a comparison in terms of GFLOPs (obtained via fvcore package at $384\times384$ input resolution) and learnable parameters between previous state-of-the-arts and ours. It can be seen that the parameter amount of our model is comparable to that of RAGNet and DSRNet, but our model is of the lowest computational complexity among all the compared methods. This efficiency is achieved by our multi-scale and non-recurrent model design. Moreover, our method demonstrates the best average performance across all datasets, as evidenced by the average numerical metrics on all testing images from Real20 and SIR^2.
>
> |Models|ERRNet|IBCLN|RAGNet|YTMT|Dong _et al._|DSRNet|Ours|
> |:-:|:-:|:-:|:-:|:-:|:-:|:-:|:-:|
> |Parameters (M)|18.58|24.37|146.72|76.90|**10.93**|124.60|131.76|
> |GFLOPs|820|708|610|838|666|743|**517**|
> |Performance|23.53/0.879|24.10/0.879|24.90/0.886|24.05/0.886|24.21/0.897|25.75/0.910|**26.49/0.922**|
> |
>
> Q4: Some parts in Figure 2 overlap excessively.
>
> A4: We will definitely polish the figures in our revised version, for better visual appeal.
>
> Q5: How does the model perform without using a pre-trained model?
>
> A5: As shown in the table below, DSIT (w/o Pretrain) refers to our model with the Transformer backbone randomly initialized, and trained from scratch alongside the main network. The results reflect a remarkable performance drop, which comes from the data-hungry nature of Transformer architectures; training from scratch on a relatively small dataset can negatively impact the performance of models [3]. This outcome also underscores the importance of high-semantic pretraining in addressing the ill-posed nature of the reflection separation task, which has been demonstrated in previous methods like the HyperColumn in YTMT [1] and DSFNet in DSRNet [2]. In future work, improvements of the pre-training technique, such as incorporating same-task pre-training strategies like HAT [4], could further enhance the model performance.
>
> Q6: Why does DSLP not use pre-trained models?
>
> A6: We have already employed the pretrained Global Prior Extractor to extract high-semantic prior, the information flow of which is task-agnostic and requires task-specific guidance to adapt to our current task. The proposed dual-stream Local Prior Extractor (LPE) serves this purpose, guiding the task-agnostic information to be more relevant to reflection separation. Therefore, there is no need to further introduce a pre-trained task-agnostic LPE.
>
> To further validate this, we experimented as shown in the table below. In the DSIT with CNN Pretrain, we replaced the original LPE with a pre-trained RepVGG-B3 model. We used CAI to interact and fuse features extracted by the pre-trained models from both architectures. However, we observed that this approach not only increased the model's overall GFLOPs but also resulted in weaker performance, which confirms our opinion above that using a pre-trained CNN network in place of our LPE does not improve performance. Thank you for your insightful question.
>
> |Models|Average PSNR/SSIM|GFLOPs|
> |:-|:-:|:-:|
> |DSIT(w/o Pretrain)|25.07/0.897|517|
> |DSIT(CNN Pretrain)|25.83/0.911|525|
> |DSIT(Ours)|**26.27**/**0.917**|**517**|
> |
>
> [1] Trash or Treasure? An Interactive Dual-Stream Strategy for Single Image Reflection Separation.  NeurIPS 2021.
>
> [2] Single Image Reflection Separation via Component Synergy. ICCV 2023.
>
> [3] Training data-efficient image transformers & distillation through attention. ICML 2021.
>
> [4] Activating More Pixels in Image Super-Resolution Transformer. CVPR 2023.

---

> > ### Comment · Reviewer_k1uS · 2024-08-12
> > **After Rebuttal**
> >
> > Thanks for the rebuttal. The authors clarify the design details and compare Params/FLOPs.
> > Overall, the authors address my concerns, so I raise my score to 6.

---

> > > ### Author Response · Authors · 2024-08-12
> > >
> > > We are honored that our response helped the reviewer better understand our paper. We sincerely appreciate the thoughtful review of the reviewer and will further refine our manuscript accordingly.

---

### Official Review · Reviewer_bMoK · 2024-07-12

**Soundness:** 2
**Presentation:** 1
**Contribution:** 2
**Rating:** 5
**Confidence:** 4

**Summary:**

This paper introduces a Dual-Stream interactive transformer to tackle the single image reflection removal task. The motivation of the proposed method is based on the drawbacks of existing methods that the dual-stream methods cannot assess the effectiveness of the information flowing between the two streams. Based on this analysis, this paper proposed . The proposed method constructs the encoder and decoder in dual-stream structure and crossover the outputs of the two sub-networks. Comprehensive experiments show that the proposed method outperforms the baseline methods.

**Strengths:**

1. This paper demonstrates the proposed method consistently outperforms the baseline methods in benchmarking datasets on image reflection separation task.
2. The motivation of the proposed method is based on observations, which is one of the existing problems of the reflection separation task.

**Weaknesses:**

1. The methodology section is not described clearly. First, in line 146, the superscript symbol k on feature parameter F_T^{k-1} is neither explained in the text nor in the figures. Does this number k indicate the network is running in recurrent manner?  And in line 149, the superscript is used to indicates layer normed features. Also in Figure 1, the superscript of feature F is using l instead.  Second, I would assume F_T represents transmission features and F_R represents reflection features.  Third, in line 151, the DS-SA  and DS-CA should be explained or at least come with citations.
2. Line 155 and Figure 2, how does DSLP FNN extract local features? How are local features defined? What is the difference between the input of DSLP and the input of the Transformer block?
3. In the introduction section, this paper criticizes the existing dual-stream methods about the insufficient assessment on the information flowing between the two streams. However, I could not see the assessment of the data or any experiments analyzing the information. To make the claim of the paper valid, the authors should show evidences that the proposed method is able to make the information positively effective for reflection separation.

**Questions:**

Generally, the paper is not well written. The equations, symbols and definitions are not clearly described in the text and therefore the idea of the methodology becomes confusing. The authors are high advised to polish the presentation of the manuscript and re-submit it. In addition, the paper does not provide enough evidence to its claim. It lacks novelty in that sense.

**Limitations:**

The limitation is addressed in the appendix: the proposed method fails to deal with the regions with dominant reflections.

---

> ### Author Rebuttal · Authors · 2024-08-07
>
> First of all, we sincerely appreciate your comments, which can help us further improve the quality of our paper. We also apologize for any confusion or inconvenience caused by writing issues. Below, we address the main concerns.
>
> **Q1**: What does the superscript symbol $k$ mean?  It is not consistent with the superscript of $\mathbf{F}$ in Figure 1.
>
> **A1**: $\mathbf{F}$ denotes the feature flow in our dual-stream architecture. Specifically, $\mathbf{F_T}$ and $\mathbf{F_R}$ represent the feature flows for the transmission and reflection layers, respectively. When the superscript of $\mathbf{F}$ is a lowercase letter, such as in $\mathbf{F}^k_\mathbf{T}$, it indicates the feature flow of the transmission layer after passing through $k$ DAIBs (Dual-Attention Interactive Blocks). The superscripts $l$ in Figure 1 and $k$ in the methodology section have the same meaning.  Moreover, when the superscript is formed by uppercase letters, such as in $\mathbf{F}^{\text{LN}}_\mathbf{T}$, it designates the feature flow after passing through a specific layer within a DAIB (in this case, LayerNorm). We appreciate your attention to this detail. We will definitely unify the notations in the next version for clarity.
>
> **Q2**: The DS-SA and DS-CA should be explained or at least come with citations.
>
> **A2**: The DS-SA (Dual-Stream Self-Attention) and DS-CA (Dual-Stream Cross-Attention) mechanisms are introduced in Section 3.1 of our paper, which are explained in the subsequent paragraphs following their first mention. In the revision, we will ensure that a brief explanation is provided at their first mention to clarify their roles as follows:
>
> "Following the LayerNorm, we apply Dual-Stream Self-Attention (DS-SA) to $\mathbf{X}^{\text{LN}}\_{0}$ and Dual-Stream Cross-Attention (DS-CA) to $\mathbf{X}^{\text{LN}}\_{1}$, obtaining $\mathbf{X}\_{\text{SA}}$ and $\mathbf{X}\_{\text{CA}}$, respectively. The design of these two attention mechanisms will be detailed later in this section."
>
> **Q3**: (1) How does DSLP FFN extract local features? (2) How are local features defined?  (3) What is the difference between the input of DSLP and the input of the Transformer block?
>
> **A3**: (1) Extraction of local features by DSLP FFN: The DSLP FFN (Dual-Stream Locality-Preserving Feed-Forward Network) and DSLP Block can be any convolutional dual-stream network modules, such as the YTMT Block [1]  and MuGI Block [2]. In our experiments, we employ the MuGI Block. Since reflection decomposition is a dense prediction task, one of the primary functions of the DSLP is to maintain local information. DSLP achieves this goal by using convolutional structures to extract local features, like other hybrid Transformer architectures [4],[5],[6].
>
> (2) Definition of local features: Local features refer to the local correlation among adjacent pixels that often have similar values. CNNs capture the local structures through limited receptive fields, weight sharing, and/or spatial sub-sampling [3],[4].
>
> (3) Differences between inputs to DSLP and Transformer blocks: Both the features for the Transformer block and the DSLP block originate from the input image $\mathbf{I}$. However, the features fed into the Transformer block first undergo a PatchEmbed process. Before being input into the DSLP FFN, these features are reshaped after the LayerNorm to fit the convolution operation, like other hybrid Transformer architectures [5],[6].
>
> To avoid any confusion, we will provide a detailed explanation in the next version to address these mentioned points thoroughly.
>
> **Q4**: How does the proposed method assess the information flow between the two streams?
>
> **A4**: We explain in detail how explicit correlation assessment works in the Common Issue in the Global Rebuttal. Additionally, the ablation study on DAIB in Table 3 of the main body of our paper shows that removing the cross-attention mechanism ("w/o DS-CA") leads to significant performance degradation on real-world datasets. This also demonstrates that our proposed method is able to make the information positively effective for reflection separation.
>
> **Q5**: Does the proposed method fail to deal with the regions with dominant reflections?
>
> **A5**: In the situation mentioned by the reviewer (Figure 13), some regions in the image are dominant with strong reflections (in other words, information of the transmission layer has been largely suppressed), which poses challenges not only to our method but also to previous state-of-the-arts. While other methods struggle to discriminate the reflection component from the entangled layers in such scenarios, our method removes significant portions of the reflections, way more effectively, which corroborates our generalization capability.
>
> [1] Trash or Treasure? An Interactive Dual-Stream Strategy for Single Image Reflection Separation.  NeurIPS 2021.
>
> [2] Single Image Reflection Separation via Component Synergy. ICCV 2023.
>
> [3] Object recognition with gradient-based learning. CGCV 1999.
>
> [4] CvT: Introducing Convolutions to Vision Transformers. ICCV 2021.
>
> [5] Incorporating Convolution Designs into Visual Transformers.  ICCV 2021.
>
> [6] Uformer: A General U-Shaped Transformer for Image Restoration. CVPR 2022.

---

> ### Comment · Reviewer_bMoK · 2024-08-13
> **Comments on rebuttal**
>
> The authors' rebuttal works have addressed most of the weaknesses raised in the initial review. Generally, the main concern is addressed and I've raised the score to 5.

---

> > ### Author Response · Authors · 2024-08-13
> >
> > Thank you for your professional review, which has been instrumental in helping us improve our paper. We will further enhance the presentation of our revised version to eliminate the writing problems.

---

### Author Rebuttal · Authors · 2024-08-07

**Common Issue**:

**Q**: Illustration of  Explicit Correlation Assessment and the Cross-Attention mechanisms in the proposed method.

**A**: We utilize the Layer-aware Cross-Attention (LaCA) mechanism for explicit correlation assessment between the two streams. A formal derivation of LaCA has already been provided in Appendix A.2 of our initial submission. We here again explain its principle as follows:

Given the feature streams of transmission layer $\textbf{T} \in \mathbb{R}^{N\times C}$ and reflection layer $\textbf{R} \in \mathbb{R}^{N\times C}$ , we concatenate them along the first dimension to form a matrix $\textbf{X} \in \mathbb{R}^{2N\times C} = \begin{bmatrix} \textbf{T} \\\ \textbf{R} \end{bmatrix}$, where $N$ represents the number of tokens and $C$ denotes the number of channels in each token, respectively.

The query $\mathbf{Q} \in \mathbb{R}^{N\times D}$, key $\textbf{K} \in \mathbb{R}^{N\times D}$, and value $\textbf{V} \in \mathbb{R}^{N\times D}$ matrices can be computed by applying the linear projections to the merged stream $\mathbf{X}$ via:

$$\textbf{Q} = \textbf{X}\textbf{W}_q , \quad \textbf{K} = \textbf{X}\textbf{W}_k, \quad \textbf{V} = \textbf{X}\textbf{W}_v ,$$
where $\textbf{W}_q, \textbf{W}_k, \textbf{W}_v \in \mathbb{R}^{C \times D} $ are the weight matrices of linear projections, changing the number of channels of each token from $C$ to a hidden dimension $D$.

The attention score matrix $\mathbf{A}\in\mathbb{R}^{2N\times 2N}$ is computed by:
$$
\textbf{A} = \text{Softmax}(\frac{\textbf{Q}\textbf{K}^T}{\sqrt{D}}) = \text{Softmax}(\frac{1}{\sqrt{D}}\begin{bmatrix}
 \textbf{T} \\\\
 \textbf{R}
\end{bmatrix}\textbf{W}_q\textbf{W}_k^T\begin{bmatrix}
 \textbf{T}^T & \textbf{R}^T
\end{bmatrix}) = \text{Softmax}( \frac{1}{\sqrt{D}}\begin{bmatrix}
 \textbf{T}\textbf{W}_q\textbf{W}_k^T\textbf{T}^T & \textbf{T}\textbf{W}_q\textbf{W}_k^T\textbf{R}^T \\\\
 \textbf{R}\textbf{W}_q\textbf{W}_k^T\textbf{T}^T & \textbf{R}\textbf{W}_q\textbf{W}_k^T\textbf{R}^T
\end{bmatrix} ),$$
where the intra-layer terms $\textbf{T}\textbf{W}_q\textbf{W}_k^T\textbf{T}^T$and $\textbf{R}\textbf{W}_q\textbf{W}_k^T\textbf{R}^T$ represent interactions within the transmission stream $\textbf{T}$ and the reflection stream $\textbf{R}$, respectively.  The inter-layer terms $\textbf{T}\textbf{W}_q\textbf{W}_k^T\textbf{R}^T$and $\textbf{R}\textbf{W}_q\textbf{W}_k^T\textbf{T}^T$ indicate interactions between the transmission stream $\textbf{T}$ and the reflection stream $\textbf{R}$.

By denoting the Softmax function with a scaling factor $\frac{1}{\sqrt{D}}$ as $\mathcal{S}(\cdot)$, the output matrix $\textbf{Y}$ is then calculated as:

$$\textbf{Y} = \textbf{A}\textbf{V} = \begin{bmatrix}
 \mathcal{S}(\textbf{T}\textbf{W}_q\textbf{W}_k^T\textbf{T}^T) \textbf{T} \textbf{W}_v + \mathcal{S}(\textbf{T}\textbf{W}_q\textbf{W}_k^T\textbf{R}^T) \textbf{R} \textbf{W}_v \\\\
 \mathcal{S}(\textbf{R}\textbf{W}_q\textbf{W}_k^T\textbf{T}^T) \textbf{T} \textbf{W}_v + \mathcal{S}(\textbf{R}\textbf{W}_q\textbf{W}_k^T\textbf{R}^T) \textbf{R} \textbf{W}_v
\end{bmatrix}.$$

We further simplify the form of $\textbf{Y}$ by introducing a function $\mathcal{G}(\textbf{A},\textbf{B})=\mathcal{S}(\textbf{A}\textbf{W}_q\textbf{W}_k^T\textbf{B}^T)\textbf{B}\textbf{W}_v$, where $\mathbf{A}\in\mathbb{R}^{N\times C}$ and $\mathbf{B}\in\mathbb{R}^{N\times C}$ can be chosen between $\mathbf{T}$ and $\mathbf{R}$, yielding the follows:

$\textbf{Y} = \begin{bmatrix}
\mathcal{G}(\textbf{T},\textbf{T})+\mathcal{G}(\textbf{T},\textbf{R}) \\\\
\mathcal{G}(\textbf{R},\textbf{T})+\mathcal{G}(\textbf{R},\textbf{R})
\end{bmatrix}.$

We finally obtain the output of the dual streams as $\textbf{T}_o=\mathcal{G}(\textbf{T},\textbf{T})+\mathcal{G}(\textbf{T},\textbf{R})$ and $\textbf{R}_o=\mathcal{G}(\textbf{R},\textbf{R})+\mathcal{G}(\textbf{R},\textbf{T})$.
Specifically, the output features of the transmission stream, $\mathbf{T}_o$, consist of two parts: intra-layer explicit correlation assessment $\mathcal{G}(\mathbf{T},\mathbf{T})$ and inter-layer explicit correlation assessment $\mathcal{G}(\mathbf{T},\mathbf{R})$. Similarly, the output features of the reflection stream, $\mathbf{R}_o$, include intra-layer explicit correlation assessment $\mathcal{G}(\mathbf{R},\mathbf{R})$ and inter-layer explicit correlation assessment $\mathcal{G}(\mathbf{R},\mathbf{T})$.

Additionally, to facilitate a more intuitive understanding of LaCA, we have included an illustrative example in Figure 3 of the **Global Rebuttal PDF** (attachment below), which sequentially displays the transmission stream $\mathbf{T}$, the reflection stream $\mathbf{R}$, the concatenated matrix along the token channel $\textbf{X}$, and 4 internal blocks of the attention matrix $\mathbf{A}^* = \begin{bmatrix}
\textbf{T}\textbf{W}_q\textbf{W}_k^T\textbf{T}^T & \textbf{T}\textbf{W}_q\textbf{W}_k^T\textbf{R}^T \\\\
 \textbf{R}\textbf{W}_q\textbf{W}_k^T\textbf{T}^T & \textbf{R}\textbf{W}_q\textbf{W}_k^T\textbf{R}^T
\end{bmatrix}$.

*The diagram clearly shows that the two submatrices along the main diagonal represent the self-attention of the layers, while the other two along the off-diagonal represent the cross-attention of the two layers*.

---

> ### Comment · Area_Chair_STTT · 2024-08-08
>
> The markdown equation display issue is resolved. I can see the equations after refreshing my browser. Please ignore my last message.
>
> -AC

---

### Decision · Program_Chairs · 2024-09-25

**Decision:**

Accept (poster)

**Comment:**

The reviewers acknowledge the technical contribution of DAI mechanism and the strong performance advantage of this paper. The rebuttal was successful, resulting in an increased average score, with previous negative scores now becoming supportive. After careful discussion and consideration, we are pleased to inform you that your paper has been accepted. There is one issue from the AC-reviewer discussion regarding the distinction in the technical aspects compared to YTMT [22]. The ACs have investigated this matter and found the technical components to be distinguishable. Moreover, the performance of this submission is significantly superior to YTMT, both quantitatively on benchmark and visually in real cases. The authors are encouraged to provide a clearer distinction in the final version. Additionally, a common suggestion from the reviewers is that the writing can be improved. While the ACs believe this will not influence the acceptance of the paper, given its merits and strong performance, the authors are advised to revise the writing substantially to enhance clarity.